# Klf5 down-regulation induces vascular senescence through eIF5a depletion and mitochondrial fission

Dong Ma[1,2], Bin Zheng[1], He-liang Liu[2], Yong-bo Zhao[3], Xiao Liu[3], Xin-hua Zhang[1], Qiang Li[1], Wei-bo Shi[4], Toru Suzuki[5], Jin-kun Wen[1]*

1 Department of Biochemistry and Molecular Biology, Key Laboratory of Neural and Vascular Biology, Ministry of Education, Hebei Medical University, Shijiazhuang, China, 2 School of Public Health, North China University of Science and Technology, Tangshan, China, 3 Department of Cardiac surgery, the Fourth Hospital of Hebei Medical University, Shi Jiazhuang, China, 4 Department of Forensic Medicine, Hebei Medical University, Hebei Key Laboratory of Forensic Medicine, Collaborative Innovation Center of Forensic Medical Molecular Identification, Shijiazhuang, China, 5 Department of Cardiovascular Sciences, University of Leicester, Leicester, United Kingdom

* wjk@hebmu.edu.cn

**Data Availability Statement:** All relevant data are within the paper and its Supporting Information files. All data of microarray and RNA sequencing used in this paper are from NCBI, and the

## Abstract

Although dysregulation of mitochondrial dynamics has been linked to cellular senescence, which contributes to advanced age-related disorders, it is unclear how Krüppel-like factor 5 (Klf5), an essential transcriptional factor of cardiovascular remodeling, mediates the link between mitochondrial dynamics and vascular smooth muscle cell (VSMC) senescence. Here, we show that Klf5 down-regulation in VSMCs is correlated with rupture of abdominal aortic aneurysm (AAA), an age-related vascular disease. Mice lacking Klf5 in VSMCs exacerbate vascular senescence and progression of angiotensin II (Ang II)–induced AAA by facilitating reactive oxygen species (ROS) formation. Klf5 knockdown enhances, while Klf5 overexpression suppresses mitochondrial fission. Mechanistically, Klf5 activates eukaryotic translation initiation factor 5a (eIF5a) transcription through binding to the promoter of eIF5a, which in turn preserves mitochondrial integrity by interacting with mitofusin 1 (Mfn1). Accordingly, decreased expression of eIF5a elicited by Klf5 down-regulation leads to mitochondrial fission and excessive ROS production. Inhibition of mitochondrial fission decreases ROS production and VSMC senescence. Our studies provide a potential therapeutic target for age-related vascular disorders.

## Introduction

Cellular senescence is an important contributor to aging and age-related diseases, and the accumulation of cellular senescence is a main feature of aged organisms [1]. Cellular senescence is traditionally defined as permanent cell cycle arrest in response to different damaging stimuli [1,2]. In the cardiovascular system, the senescence of vascular smooth muscle cells (VSMCs) may be induced by different stimuli, such as angiotensin II (Ang II), oxidative stress, inflammation, and DNA damage. On the other hand, VSMC senescence may also results in

accession numbers from NCBI are GSE148841 and GSE148765, respectively. Data used for figure generation are provided in the Figures.data and supplemental Figures.data Excel sheets. All images of blot and gel are available from the PDF file.

**Funding:** J-KW received funding from the National Natural Science Foundation of China (No. 31671182 and No. 31871152) and Hebei scientific research project of high level talents (GCC2014026). DM is funded by the National Natural Science Foundation of China (No. 81700416). The funders had no role in study design, data collection and analysis, decision to publish, or preparation of the manuscript.

**Competing interests:** The authors have declared that no competing interests exist.

**Abbreviations:** AAA, abdominal aortic aneurysm; Ad-Ctl, adenoviruses encoding control; Ad-Klf5, adenoviruses encoding Klf5; Ad-shKlf5, adenoviruses encoding small hairpin Klf5; Ad-U6, adenoviruses encoding U6; Ang II, angiotensin II; Atp5b, ATP synthase subunit β; Cdkn1a, cyclin dependent kinase inhibitor 1a; Cox4i1, cytochrome c oxidase subunit 4i1; Cox6a2, mitochondrial cytochrome c oxidase subunit 6A isoform 2; CT, computed tomography; DHE, dihydroethidium; DMEM, Dulbecco's Modified Eagle Medium; Drp1, dynamin-related protein 1; EC, endothelial cell; eIF5a, eukaryotic translation initiation factor 5a; ERK, extracellular signal–regulated kinase; FACS, flow analysis of cytosorting; Fis1, fission mitochondrial 1; GAPDH, glyceraldehyde-3-phosphate dehydrogenase; GO, gene ontology; GPCR, G protein–coupled receptor; HE, hematoxylin–eosin; IgG, immunoglobulin G; KEGG, Kyoto Encyclopedia of Genes and Genomes; Klf5, Krüppel-like factor 5; Mapk14, mitogen-activated protein kinase 14; MCP-1, monocyte chemoattractant protein 1; Mdivi-1, mitochondrial division inhibitor 1; Mfn1, mitofusin 1; Mfn2, mitofusin 2; MMP2, matrix metalloproteinase 2; mtDNA, mitochondrial DNA; Mtfr1, mitochondrial fission regulator 1; MTG, MitoTracker Green; mtROS, mitochondrial ROS; mtTFA, mitochondrial transcription factor A; NAC, N-acetyl-L-cysteine; Nfe2l2, nuclear factor, erythroid 2 like 2; NOX, NAPDH oxidase; PDGF, platelet-derived growth factor; PGC1α, peroxisome proliferative activated receptor, gamma, coactivator 1 alpha; Pink1, PTEN-induced kinase 1; PPARα, peroxisome proliferator-activated receptor α; pRL-TK, thymidine kinase promoter-Renilla luciferase reporter plasmid; qRT-PCR, quantitative real-time PCR; RNA-Seq, RNA sequencing; ROS, reactive oxygen species; SA-β-gal, senescence-associated β-galactosidase; Si-Ctl, short interfering RNA

the loss of arterial function, chronic vascular inflammation, mitochondrial dysfunction, and the development of age-related vascular disorders, such as atherosclerosis, abdominal aortic aneurysm (AAA), hypertension, and diabetes [3,4].

Krüppel-like factor 5 (Klf5) is a zinc-finger transcriptional factor that regulates various cellular processes, including proliferation, differentiation, development, and apoptosis [5]. In VSMCs, Klf5 is regulated by Ang II signaling and is an essential regulator of cardiovascular remodeling [6]. Ang II is known not only to regulate blood pressure and electrolyte balance but also to be involved in mediation of cell proliferation and oxidative stress, thus contributing to premature senescence [7,8]. During cardiovascular remodeling, Klf5 activates the expression of cell cycle promoting genes, such as cyclin D1, cyclin B1, and growth factors and their receptors, such as platelet-derived growth factor (PDGF), vascular endothelial growth factor (VEGF), and VEGF receptor [9,10], with a concomitant suppression of negative cell cycle control gene p21 [11,12]. Our recent studies demonstrate that Klf5 is highly expressed in both macrophages and VSMCs of human and experimental mouse AAA. Moreover, Klf5 expression progressively increases with the aortic diameter expansion during Ang II infusion–induced AAA formation [13]. Despite the facts that (1) advanced age is a major risk factor for AAAs [14], (2) VSMC senescence correlates with increased level of reactive oxygen species (ROS) [15], and (3) Klf5 participates in development and progression of AAAs [13], it remains to be clarified whether and how Klf5 mediates the mechanistic and functional link between ROS generation and VSMC senescence.

Mitochondria play important roles in regulating critical cellular, physiological, and pathophysiological processes [16,17]. Alterations in mitochondrial dynamics, which is regulated mainly by the processes of fission and fusion, are implicated in various human diseases, including cancer and neurologic and cardiovascular diseases [17–19]. Previous studies have demonstrated that dysregulation of mitochondrial dynamics is a key feature of aging [17]. Moreover, mitochondria are a major source of ROS, and mitochondrial dysfunction may lead to aberrant ROS production [20]. Specifically, elevated ROS levels impair vascular cell life span through the onset of cellular senescence [7] and have been demonstrated in human cerebral aneurysm [21]. These results indicate that dysregulation of mitochondrial dynamics can drive VSMC senescence and excessive ROS production. Although cardiomyocyte Klf5 was recently identified as a regulator of cardiac metabolism by directly activating transcription of peroxisome proliferator-activated receptor α (PPARα) and regulating lipid metabolism [22], whether and how Klf5 regulates mitochondrial dynamics is currently unclear.

It is well known that the dynamin-like GTPases mitofusin 1 and 2 (Mfn1, Mfn2) regulate the mitochondrial fusion process, and dynamin-related protein 1 (Drp1) plays a central role in the regulation of mitochondrial fission [23]. Eukaryotic translation initiation factor 5a (eIF5a) is localized not only to the nucleus but also to the mitochondria [24] and is involved in the regulation of redox homeostasis [25]. However, it remains unclear whether eIF5a, together with mitochondrial dynamics–related proteins, participates in the regulation of mitochondrial dynamics. Therefore, it is very intriguing to investigate whether eIF5a mediates functional link between Klf5 and mitochondrial dynamics, as well as to explore the relationship between Klf5-regulated mitochondrial dynamics and VSMC senescence.

## Results

### Down-regulation of Klf5 expression in VSMCs is correlated with the progression and rupture of aortic aneurysm

Because Klf5 is well known to mediate Ang II–induced vascular remodeling by stimulating VSMC proliferation [5], we sought to know how medial VSMCs were lost in Ang II–induced

control; Si-Drp1, short interfering RNA targeting Drp1; Si-eIF5a, short interfering RNA targeting eIF5a; Si-Mfn1, short interfering RNA targeting Mfn1; siRNA, small interfering RNA; SMA, smooth muscle α-actin; Tmx2, thioredoxin-related transmembrane protein 2; VEGF, vascular endothelial growth factor; VSMC, vascular smooth muscle cell; XPO1, nuclear export protein exportin 1; WT, wild-type; ΔΨm, mitochondrial membrane potential.

AAA. Thus, we determined the expression of Klf5 in human unruptured and ruptured AAAs. Demographic characteristics and representative three-dimensional volume-rendered images from axial computed tomography (CT) scans as well as histological analyses with hematoxylin–eosin (HE) and Masson's trichrome staining are presented in **Table 1** and **S1A–S1C Fig**. And the results of western blot showed that Klf5 expression in ruptured AAA (4 cases) was significantly lower than that in unruptured AAA (22 cases), even though its expression was higher than that in normal abdominal aorta (8 cases) (**Fig 1A and 1B**). Furthermore, confocal microscopy images were obtained by immunofluorescence staining with VSMC marker (smooth muscle α-actin, SMA) and Klf5, and showed that Klf5 was co-localized with VSMC marker in unruptured AAA (**Fig 1C**). Moreover, Klf5-positive VSMCs were hardly observed in ruptured AAA, while the percentages of Klf5-positive VSMCs were significantly higher in unruptured AAA than in the normal aorta (10.3% ± 2.2% versus 2.4% ± 0.89%; **Fig 1C and 1D**).

In further studies, AAA models were generated in ApoE$^{-/-}$ mice by chronic infusion of Ang II for 28 or 42 days, showing a typical aneurysmal phenotype or aneurysm rupture (**Fig 1E**). The aortic external diameter of Ang II–infused mice for 28 and 42 days increased obviously compared with that of the control mice (**Fig 1F**). Histological analysis with hematoxylin-eosin and Masson's trichrome staining revealed an obvious collagen deposition in the medial layer of unruptured AAA (28 days) or collagen breakdown in ruptured AAA (42 days) (**Fig 1G**). We also determined the expression of Klf5 in VSMCs of AAA models and found that Klf5 expression and Klf5-positive VSMCs were significantly increased at 14 and 28 days

**Table 1. Demographic and clinical characteristics of patients with AAA.**

| Item | Patients with AAA (*n* = 22) | Patients with ruptured AAA (*n* = 4) |
|---|---|---|
| General characteristics | | |
| Age, years | 71 (65–78) | 77 (72–82) |
| Cardiovascular risk factors | | |
| Diabetes, *n* (%) | 6 (27.3%) | 1 (25.0%) |
| Arterial hypertension, *n* (%) | 16 (72.3%) | 3 (75.0%) |
| Dyslipidemia, *n* (%) | 12 (54.5%) | 4 (100%) |
| Smoking, *n* (%) | 11 (50.0%) | 3 (75.0%) |
| Clinical characteristics | | |
| Asymptomatic AAA, *n* (%) | 15 (68.2%) | 0 (0) |
| Pain, *n* (%) | 4 (26.9%) | 4 (100%) |
| Imaging characteristics | | |
| AAA diameter, mm | 4.8 (3.1–5.7) | 7.9 (7.1–8.9) |
| Comorbidities | | |
| Cardiac failure, *n* (%) | 4 (18.2%) | 1 (25.0%) |
| Renal failure, *n* (%) | 1 (4.5%) | 1 (25.0%) |
| Chronic obstructive pulmonary disease, *n* (%) | 4 (18.2%) | 2 (50.0%) |
| Preoperative full blood count | | |
| Red blood cells ($10^{12}$/L) | 4.3 (3.7–4.5) | 3.8 (3.4–4.1) |
| Hemoglobin, g/dL | 12.9 (10.7–14.6) | 11.7 (10.1–13.2) |
| Thrombocytes ($10^9$/L) | 215 (163–279) | 191 (159–249) |
| Leukocytes ($10^9$/L) | 8.4 (7.3–10.8) | 10.2 (8.4–10.8) |
| Neutrophils ($10^9$/L) | 6.1 (4.5–8.1) | 5.6 (4.5–7.3) |
| Lymphocytes ($10^9$/L) | 1.6 (1.1–2.1) | 2.1 (1.8–2.4) |

For numerical raw data, please see S1 Data.

Abbreviation: AAA, abdominal aortic aneurysm

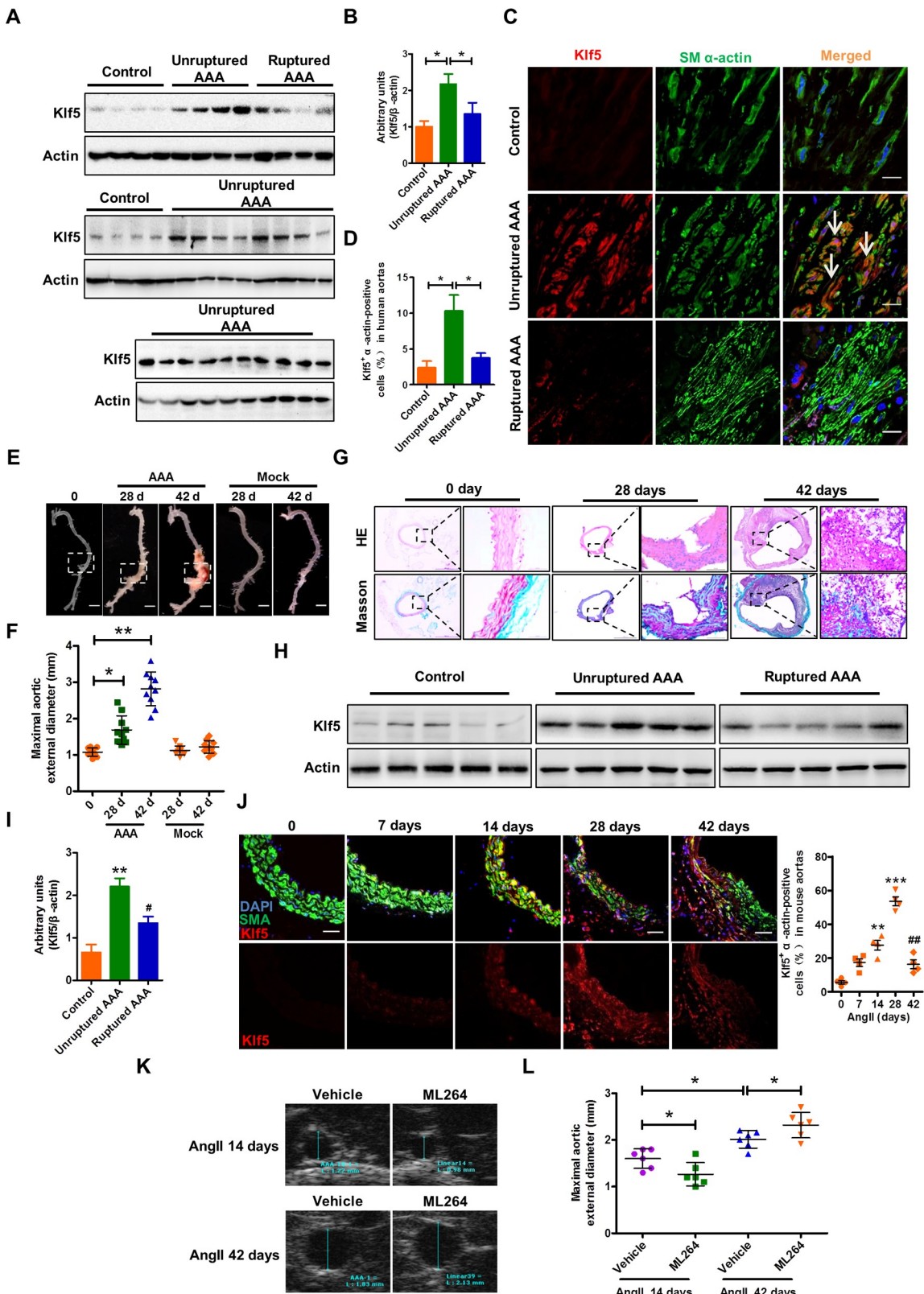

**Fig 1. Klf5 expression is down-regulated in the ruptured human and experimental AAA.** (A) The images of all western blots of Klf5 in human unruptured ($n = 22$) and ruptured ($n = 4$) AAA samples and in control aortic tissues ($n = 8$). (B) Band intensities that were

measured and normalized to β-actin. $^{*}P < 0.05$ versus control or unruptured AAA. (C) Tissue sections were prepared from frozen samples from the unruptured and ruptured AAA patients same as those in (A), and immunofluorescence staining of SM α-actin, Klf5, and DAPI was performed. Arrows indicate Klf5$^{+}$ SM α-actin-positive cells. Scale bars = 50 μm. (D) The percentage of Klf5$^{+}$ SM α-actin-positive cells. $^{*}P < 0.05$ vs. control or unruptured AAA. (E) Representative photographs of aortas from ApoE$^{−/−}$ mice infused with Ang II (1000 ng/kg/min) or saline (Mock) for 0, 28 and 42 days. Scale bars = 2.5 mm. (F) Statistical analysis of the maximal aortic external diameter. n = 10 for each group. Data represent the mean ± SD. $^{*}P < 0.05$ and $^{**}P < 0.01$ vs. 0 day. (G) Hematoxylin-eosin (top row) and Masson trichrome staining (bottom row) of abdominal aortic sections from ApoE$^{−/−}$ mice treated as in (E). (H) Western blot analysis detected Klf5 expression in mouse unruptured and ruptured AAA induced by Ang II in ApoE$^{−/−}$ mice as well as in the aortas from control mice infused with saline (n = 5 for each group). (I) Band intensities that were measured and normalized to β-actin. $^{**}P < 0.01$ vs. control; $^{#}P < 0.01$ vs. unruptured AAA. (J) Immunofluorescence staining of SM α-actin (SMA) and Klf5 in the injured aortas of ApoE$^{−/−}$ mice exposed to Ang II for the indicated days. Scale bars = 50 μm. Right: Statistical analysis of Klf5$^{+}$ SM α-actin–positive cells. $^{**}P < 0.01$ and $^{***}P < 0.001$ versus 0 day; $^{##}P < 0.01$ versus 28 days. n = 4 for each time point. (K) Representative ultrasound imaging of mouse AAA models induced by Ang II infusion for 14 and 42 days in ApoE$^{−/−}$ mice injected intraperitoneally with ML264 every two days for 14 and 42 days. (L) Statistical analysis of the maximal aortic external diameter. Data represent the mean ± SEM. $^{*}P < 0.05$ versus vehicle. n = 6 for each group. For numerical raw data, please see S1 Data. For raw immunoblots, please see S1 Blots. AAA, abdominal aortic aneurysm; Ang II, angiotensin II; Klf5, Krüppel-like factor 5; SM, smooth muscle.

after infusion with Ang II and subsequently started to decrease by 42 days (Fig 1J). Moreover, Klf5 expression level also was lower in ruptured AAA than in unruptured AAA (Fig 1H).

To identify that Klf5 expression is either beneficial or detrimental to AAA development and progression, we pharmacologically suppressed the expression of Klf5 by injecting intraperitoneally ML264 into mice infused with Ang II for 14 days (early stage) and 42 days (early+late stage). Ultrasound imaging showed that ML264 significantly suppressed Ang II–induced aortic dilation at the early stage compared with the vehicle control, whereas it accelerated Ang II–induced aortic dilation upon continuous administration by 42 days (early+late stage) (Fig 1K and 1L). Consistently, HE staining showed that ML264 treatment reduced aortic luminal dilation and wall thinning in Ang II–infused mice at the early stage, while accelerating luminal dilation, aortic dissection, and wall thinning upon continuous administration by 42 days compared with the vehicle control (S2 Fig).

To clarify whether the down-regulation of Klf5 expression at the late stage of the AAA is an effect of desensitization of the Ang II receptor, we performed western blot analysis to detect extracellular signal–regulated kinase 1/2 (ERK1/2) signaling, which is one of signal pathways activated by Ang II binding to the AT1 receptor, and the expression of β-arrestin2 that mediates desensitization of G protein–coupled receptors (GPCRs) in Ang II–induced mouse AAA models. As expected, long-term treatment with Ang II (28 and 42 days) significantly increased the expression level of phosphor-ERK1/2, and β-arrestin2 expression also had a modest increase (S3B Fig). Considering our mRNA microarray analysis showing a 3.2-fold increase of β-arrestin2 expression in Ang II–injured aortas from smcKlf5$^{−/−}$ mice relative to that from wild-type (WT) mice, and that Klf5 deficiency in VSMCs facilitated β-arrestin2 expression (S3C Fig), it is reasonable to conclude that Klf5 down-regulation at the late stage of mouse AAA could lead to β-arrestin2 up-regulation through an as yet unidentified mechanism. Based on the evidence that intensity of Ang II signaling was strengthened, it can be concluded that the attenuation in Klf5 expression at the late stage of the disease could not be an effect of desensitization of the Ang II receptor. These results imply that Klf5 appears to be detrimental in the early stage of AAA formation and that Klf5 down-regulation in the late stage of AAA is correlated with the progression and rupture of aortic aneurysm.

## Smooth muscle cell-specific knockout of Klf5 (smcKlf5$^{−/−}$) exacerbates the progression of Ang II–induced AAA by facilitating ROS formation

To further investigate the VSMC-specific functions of Klf5 in the pathogenesis of AAA, we generated mice lacking Klf5 specifically in SMCs (Klf5$^{f/f}$/Sm22$^{Cre/+}$). Because advanced age is a

known risk factor for AAA formation [26], we used young (3-month-old) and old (18-month-old) Klf5$^{flox/flox}$ mice (control, equivalent to WT) or smcKlf5$^{-/-}$ mice to establish experimental AAA models by chronic Ang II infusion (**Fig 2A**). As shown in **S4A and S4B Fig**, aging led to an increase in senescence-associated β-galactosidase (SA-β-gal)–positive staining regions in the aortas of WT mice, and the Klf5 loss in VSMCs further increased the areas of SA-β-gal–positive staining. The incidence of AAA in old mice had a significant increase compared with young mice regardless of Klf5 deletion in VSMCs; Klf5 deficiency resulted in a further increase in AAA incidence (**Fig 2B**). The aortic diameter of young smcKlf5$^{-/-}$ mice had a >30% increase compared to young WT mice (1.73 ± 0.31 mm versus 1.25 ± 0.28 mm, $P < 0.05$), but there was no difference in the survival rates between young WT and smcKlf5$^{-/-}$ mice. In old mice, smcKlf5$^{-/-}$ mice also had a >30% increase in the aortic diameter compared with WT mice (2.05 ± 0.38 mm versus 1.71 ± 0.33 mm, $P < 0.05$), and the survival rates were also significantly less in smcKlf5$^{-/-}$ mice than in WT mice (75.0% versus 89.7%, $P < 0.05$; **Fig 2C and 2D**). To determine the cause of death in the smcKlf5$^{-/-}$ mice, echocardiography was used to assess the cardiac function of all the mice infused with Ang II for 4 weeks. The results showed that the left ventricular ejection fraction and shortening fraction were decreased in the smcKlf5$^{-/-}$ mice and older mice relative to young WT mice, and the decreases in these indicators were more obvious in older smcKlf5$^{-/-}$ mice (**S5A and S5B Fig**). Accordingly, there was a significant increase in dimensions of left ventricle at systole and diastole in the smcKlf5$^{-/-}$ mice and older mice relative to young WT mice (**S5C and S5D Fig**). These findings indicate that besides rupture of aortic aneurysm, impaired cardiac function was also associated with death in the smcKlf5$^{-/-}$ mice. Collectively, these results suggest that aging increases Ang II–induced AAA formation in apoE$^{-/-}$ mice and that Klf5 deficiency in VSMCs further exacerbates the progression of Ang II–induced AAA. Moreover, HE and Masson staining showed that Klf5 deficiency in VSMCs decreased collagen deposition and local thickness of media in old smcKlf5$^{-/-}$ mice compared with those of old WT mice or young smcKlf5$^{-/-}$ mice (**Fig 2E and 2F**). Notably, the extensive breakdown of collagen was observed in the ruptured aortas of old smcKlf5$^{-/-}$ mice (**Fig 2E**).

Because ROS plays a central role in the progression of AAA [27], we investigated whether loss of Klf5 in VSMCs can affect ROS production. Thus, young and old WT or smcKlf5$^{-/-}$ mice were infused chronically with Ang II, and dihydroethidium (DHE) staining (to measure total superoxide) and mitoSOX staining (to measure mitochondrial superoxide) were performed. The results showed that the levels of total ROS and mitochondrial ROS (mtROS) were significantly higher in the aortas of old mice than in the aortas of young mice, regardless of Klf5 deletion. Knockout of Klf5 in VSMCs further enhanced total ROS and the mtROS generation not only in young mice but also in old mice (**Fig 2G–2I**). These findings indicate that increased ROS production elicited by Klf5 ablation is correlated with the progression of Ang II–induced AAA.

In addition, inflammatory cell infiltration was also significantly increased in the aortas of Ang II–infused old mice regardless of Klf5 deficiency, and Klf5 depletion in VSMCs further increased their accumulation (**Fig 2J**). Correspondingly, the expression of chemokine monocyte chemoattractant protein 1 (MCP-1) and matrix metalloproteinase 2 (MMP2), which are directly related to inflammatory cell infiltration, was significantly up-regulated in the aortas of old mice, and their up-regulation was further strengthened by Klf5 knockout (**Fig 2K**). Also, TUNEL staining showed that alterations of cell apoptosis in the aortas of different mice had a similar trend to those of MCP-1 and MMP2 expression (**S6 Fig**). These results suggest that vascular senescence in old mice may actively contribute to the chronic inflammation, thus facilitating Ang II–induced AAA formation.

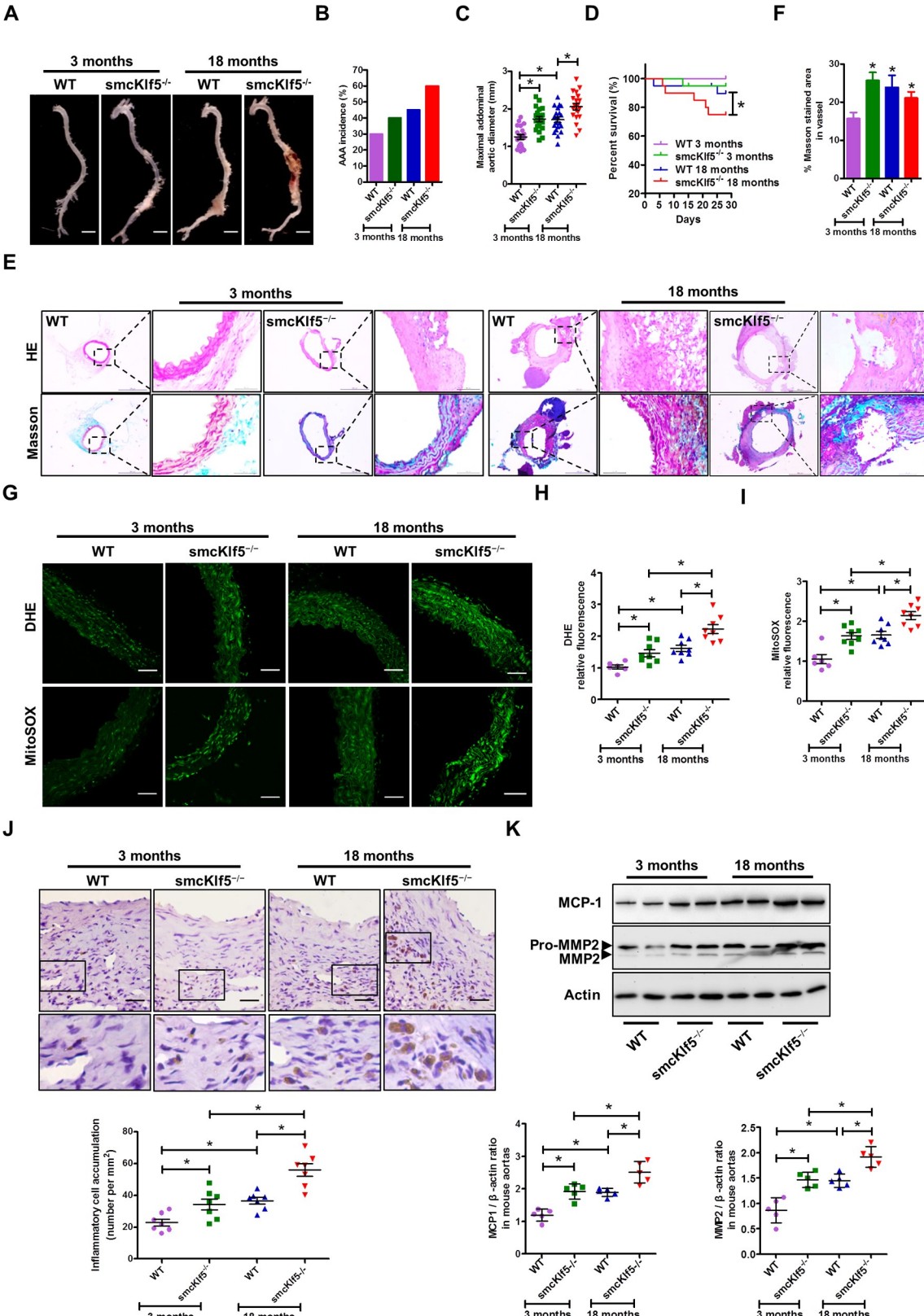

**Fig 2. SMC-specific knockout of Klf5 (smcKlf5$^{-/-}$) exacerbates the progression of Ang II–induced AAA by facilitating ROS formation.** (A) Representative photographs of aortas injured by Ang II infusion in young (3 months) or old (18 months) WT and

smcKlf5$^{-/-}$ mice. Scale bars = 2.5 mm. $n$ = 20 for each group. (B-D) Statistical analysis of the incidence (B), the maximal aortic external diameter (C), and the survival rate (D). Data represent the mean ± SD. $^*P < 0.05$ versus young or old WT mice, log-rank test for D. (E) HE-stained (top row) and Masson-stained (bottom row) sections of the aortas from young or old WT and smcKlf5$^{-/-}$ mice infused by Ang II for 28 days. Scale bars = 50 μm. (F) Quantification of Masson-stained area. $n$ = 8–10. $^*P < 0.05$ versus young WT mice. (G) The tissue sections were prepared from AAA samples of 3-month-old or 18-month-old WT and smcKlf5$^{-/-}$ mice infused with Ang II for 28 days. Dihydroethidium (DHE) (top) and mitoSOX staining (bottom) was performed to assess total cellular ROS and mitochondrial ROS. Scale bars = 50 μm. (H,I) Quantification of ROS levels based on measurement of fluorescence intensity. Data represent the mean ± SD. $^*P < 0.05$ versus WT or young smcKlf5$^{-/-}$ mice, $n$ = 8 for each group. (J) Immunostaining of CD45 on the aortic sections from young or old WT and smcKlf5$^{-/-}$ mice infused with Ang II for 28 days. Bottom: the number of CD45-positive cells accumulating in aortic wall. Data represent the mean ± SD. $^*P < 0.05$ versus WT or young smcKlf5$^{-/-}$ mice, $n$ = 7 for each group. (K) Western blotting (top) and densitometric analysis (bottom) of the protein levels of MCP-1 and MMP2. Data represent the mean ± SD. $^*P < 0.05$ versus WT or young smcKlf5$^{-/-}$ mice, $n$ = 7 for each group. For numerical raw data, please see S1 Data. For raw immunoblots, please see S1 Blots. AAA, abdominal aortic aneurysm; Ang II, angiotensin; DHE, dihydroethidium; HE, hematoxylin–eosin; Klf5, Krüppel-like factor 5; MCP-1, monocyte chemoattractant protein 1; MMP2, matrix metalloproteinase 2; ROS, reactive oxygen species; SMC, smooth muscle cell; WT, wild-type.

## Klf5 down-regulation leads to VSMC senescence

Ang II not only induces oxidative stress but also promotes VSMC senescence [28,29]. The above results indicate that SMC-specific Klf5 ablation also enhanced ROS generation. We next investigated whether Klf5 is involved in Ang II–induced ROS production and VSMC senescence. Human VSMCs were consecutively treated with Ang II (100 nmol/L) or Mock for 0, 1, 3, and 5 days, and then cellular senescence was examined by SA-β-gal staining, a classical method for detecting cellular senescence. The results showed that SA-β-gal–positive cells were significantly increased when human VSMCs were exposed to Ang II for 5 days (Fig 3A), indicating that chronic Ang II stimulation results in VSMC senescence. Because the main characteristic of senescence is cell proliferation arrest [30], we examined the expression of Ki67, a marker for proliferating cells. As a result, Ki67 levels were markedly increased 1 day after Ang II treatment but returned to the basal level by 3 days. On day 5, Ki67-positive cells were hardly observed in the cell culture (Fig 3B, S7 Fig). These data demonstrate that Ang II stimulates proliferation when acted on human VSMCs for a short time, but chronic stimulation of Ang II induces cell senescence. Furthermore, we explored the relationship between Klf5 and Ang II–induced VSMC senescence and found that Klf5 expression was significantly induced 1 day after Ang II stimulation, and then began to drop on day 3. On day 5, Klf5 expression was hardly detected by western blotting (Fig 3C), consistent with Ki67 expression changes. These findings suggest that there is a direct relationship between Klf5 down-regulation and Ang II–induced SMC senescence.

In further experiments, we used a recombinant adenovirus (Ad-shKlf5 or Ad-Klf5) to knock down or overexpress Klf5 in human VSMCs. SA-β-gal staining for VSMCs showed that the number of SA-β-gal–positive cells was dramatically reduced in Klf5-overexpressing VSMCs treated with Ang II for 5 days compared with that of adenoviruses encoding control (Ad-Ctl)–infected cells, with a concomitant decline in the total ROS and mtROS (Fig 3D–3F). Conversely, knockdown of Klf5 by Ad-shKlf5 significantly enhanced the number of aged VSMCs upon exposure to Ang II for 5 days, with a concomitant increase in the total ROS and mtROS (Fig 3D–3F). Correspondingly, Klf5 overexpression or knockdown significantly increased or attenuated the number of Ki67-positive cells in Ang II–treated VSMCs for 5 days (Fig 3G). Simultaneously, chronic stimulation of Ang II also induced apoptosis in Klf5-knocked down or overexpressed VSMCs, and Klf5 knockdown further increased the number of apoptotic cells (Fig 3H). Collectively, these results suggest that down-regulation of Klf5 facilitates VSMC senescence induced by chronic stimulation of Ang II at least in part via increasing ROS generation.

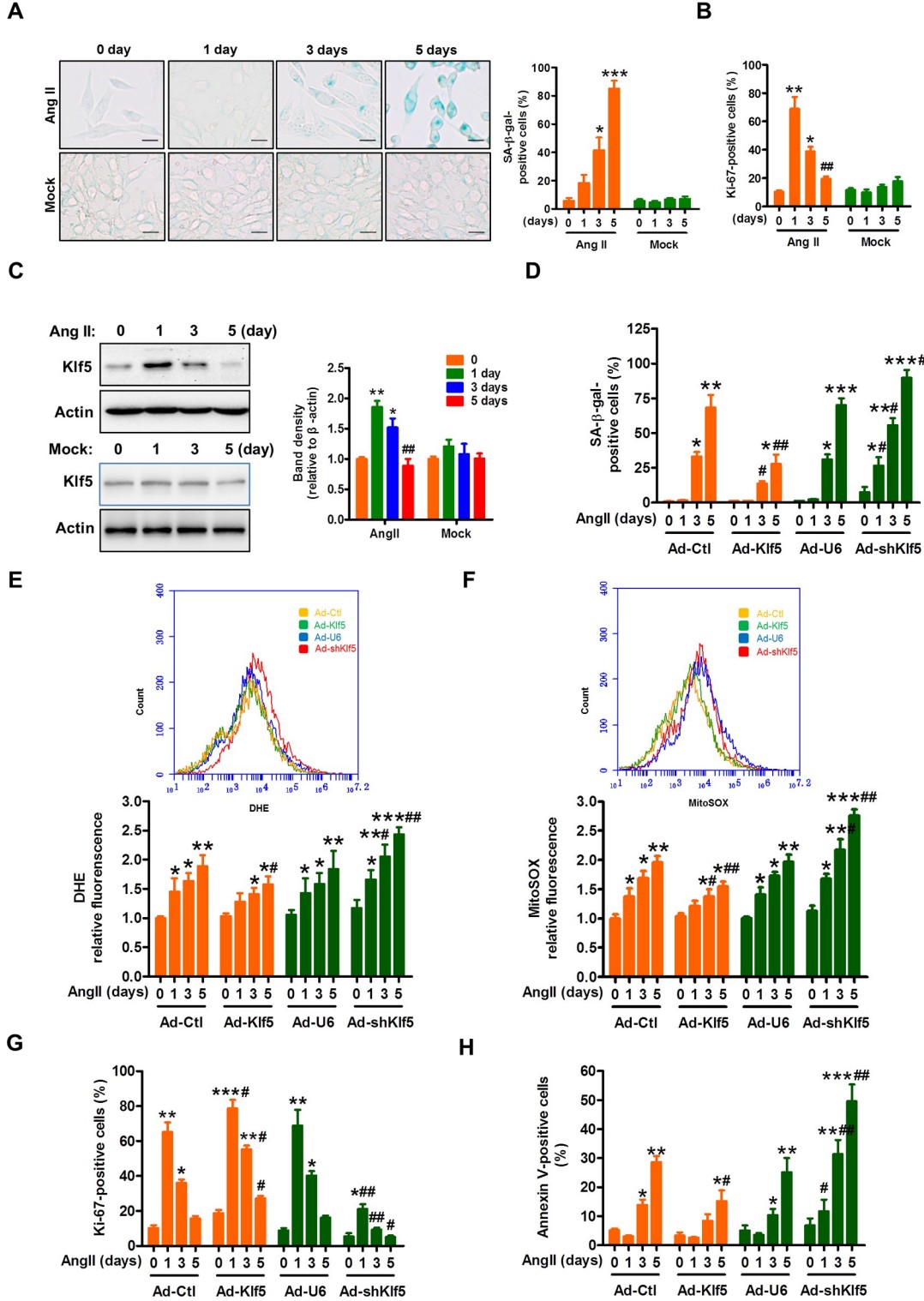

**Fig 3. Klf5 down-regulation leads to human VSMC senescence.** (A) Human VSMCs were stimulated with Ang II (100 nmol/L) or PBS as mock for the indicated times. SA-β-gal staining was performed (left), and SA-β-gal–positive cells were quantified from three different fields (right). Scale bars = 50 μm. Data represent the mean ± SEM. *$P < 0.05$ and ***$P < 0.001$ versus 0 day, $n = 3$ in each group. (B) Human VSMCs were treated as in (A), and Ki67-positive VSMCs were quantified from three different fields. Data represent the mean ± SEM. *$P < 0.05$ and **$P < 0.01$ versus 0 day, ##$P < 0.01$ versus 1 day. (C) Representative images of western blotting of Klf5 in human VSMCs treated as in (A). β-actin serves as a loading control. Band

intensities that were measured and normalized to β-actin were shown at the right side ($n$ = 3). Data represent the mean ± SEM. $^*P < 0.05$ and $^{**}P < 0.01$ versus 0 day, $^{##}P < 0.01$ versus 1 day. (D-H) Human VSMCs were infected with the indicated constructs (Ad-Ctl and Ad-Klf5 or Ad-U6 and Ad-shKlf5) and then treated with Ang II for 0, 1, 3, 5 days; SA-β-gal–positive (D), DHE-positive (E), mitoSOX-positive (F), Ki67-positive (G), and annexin V–positive (H) cells were quantified from three different fields. Data represent the mean ± SEM. $^*P < 0.05$, $^{**}P < 0.01$, $^{***}P < 0.001$ versus 0 day, $^{#}P < 0.05$ and $^{##}P < 0.01$ versus corresponding Ad-Ctl or Ad-U6 at the same day. For numerical raw data, please see S1 Data. For raw immunoblots, please see S1 Blots. Ad-Ctl, adenoviruses encoding control; Ad-shKlf5, adenoviruses encoding small hairpin Klf5; Ad-U6, adenoviruses encoding U6; Ang II, angiotensin II; DHE, dihydroethidium; Klf5, Krüppel-like factor 5; SA-β-gal, senescence-associated β-galactosidase; VSMC, vascular smooth muscle cell.

## SMC-specific ablation of Klf5 leads to mitochondrial fission

To clarify the mechanism whereby ROS production was increased in Klf5-deficient VSMCs, we first identified the genes that are regulated by Klf5 in aortic tissues. To do this, we performed RNA expression profiling using microarray analyses of the aortic tissues from smcKlf5$^{-/-}$ mice versus WT mice infused with Ang II for 28 days. As a result, 993 genes were found to be differentially expressed between smcKlf5$^{-/-}$ mice and WT mice (≥2-fold change in expression level; $P < 0.05$) (Fig 4A). Gene ontology (GO) and Kyoto Encyclopedia of Genes and Genomes (KEGG) pathway analysis revealed that down-regulated genes by Klf5 deficiency in aortic VSMCs were strongly associated with ATP production, respiratory burst, and hydrogen peroxide catabolic process (Fig 4B). To further validate these findings, we infected mouse VSMCs with Ad-Klf5 or Ad-Ctl and used RNA sequencing (RNA-Seq) analysis to compare the gene expression profiles between Ad-Klf5– and Ad-Ctl–infected VSMCs. The results showed that a total of 504 up-regulated genes (≥2-fold change in expression level; $P < 0.05$) were detected in Klf5-overexpressed VSMCs. On the basis of the GO and GSEA analysis, the up-regulated genes by Klf5, opposite to what was observed in the aortic tissues of smcKlf5$^{-/-}$ mice, mainly involved the genes related to ATP biosynthetic process and cell redox homeostasis. Notably, mitochondrial dynamics–related genes, such as Drp1, Mfn1, mitochondrial fission regulator 1 (Mtfr1), fission mitochondrial 1 (Fis1), and PTEN-induced kinase 1 (Pink1), were also significantly differentially expressed (Fig 4C). Using quantitative real-time PCR (qRT-PCR), we further validated the expression of mitochondrial dynamics–, redox homeostasis–, and ATP biosynthesis–related genes and obtained the results similar to those seen in RNA-Seq analysis (Fig 4D). Importantly, we found that eIF5a, a regulator of cell redox homeostasis [25], was also highly up-regulated in Klf5-overexpressed VSMCs, as shown by RNA-Seq analysis and qRT-PCR (Fig 4C and 4D).

Because the alteration of mitochondrial dynamics is associated with increased mtROS production [31], we focused on the effect of Klf5 knockout or overexpression on the expression of eIF5a and mitochondrial dynamics–related genes. We found that mitochondrial fission factors Drp1, Fis1, and Mtfr1 were significantly up-regulated, whereas fusion protein Mfn1 was down-regulated in the aortic VSMCs of smcKlf5$^{-/-}$ mice, without affecting the expression of several other proteins involved in the regulation of mitochondrial dynamics, as evidenced by western blot analysis (Fig 4E). In contrast, overexpression of Klf5 in mouse VSMCs produced the opposite effects on these gene expressions (Fig 4F). Furthermore, we infected Klf5-deficient VSMCs with pAd-Klf5 to assess the effect of Klf5 reintroduction on the expression of mitochondrial dynamics–related proteins. Western blotting showed that expression levels of Fis1, Drp1, Mtfr1, Mfn1, and Pink1 were restored in Klf5-reintroducing Klf5$^{-/-}$ VSMCs (S8A Fig). Meanwhile, these results also were verified in human VSMCs (S8B and S8C Fig). In addition, the rest of the differentially expressed genes identified by RNA-Seq and qRT-PCR were also verified at translation level in Klf5-deficient or overexpressed VSMCs, and changes in the expression of ATP synthase subunit β (Atp5b) and mitochondrial cytochrome c oxidase

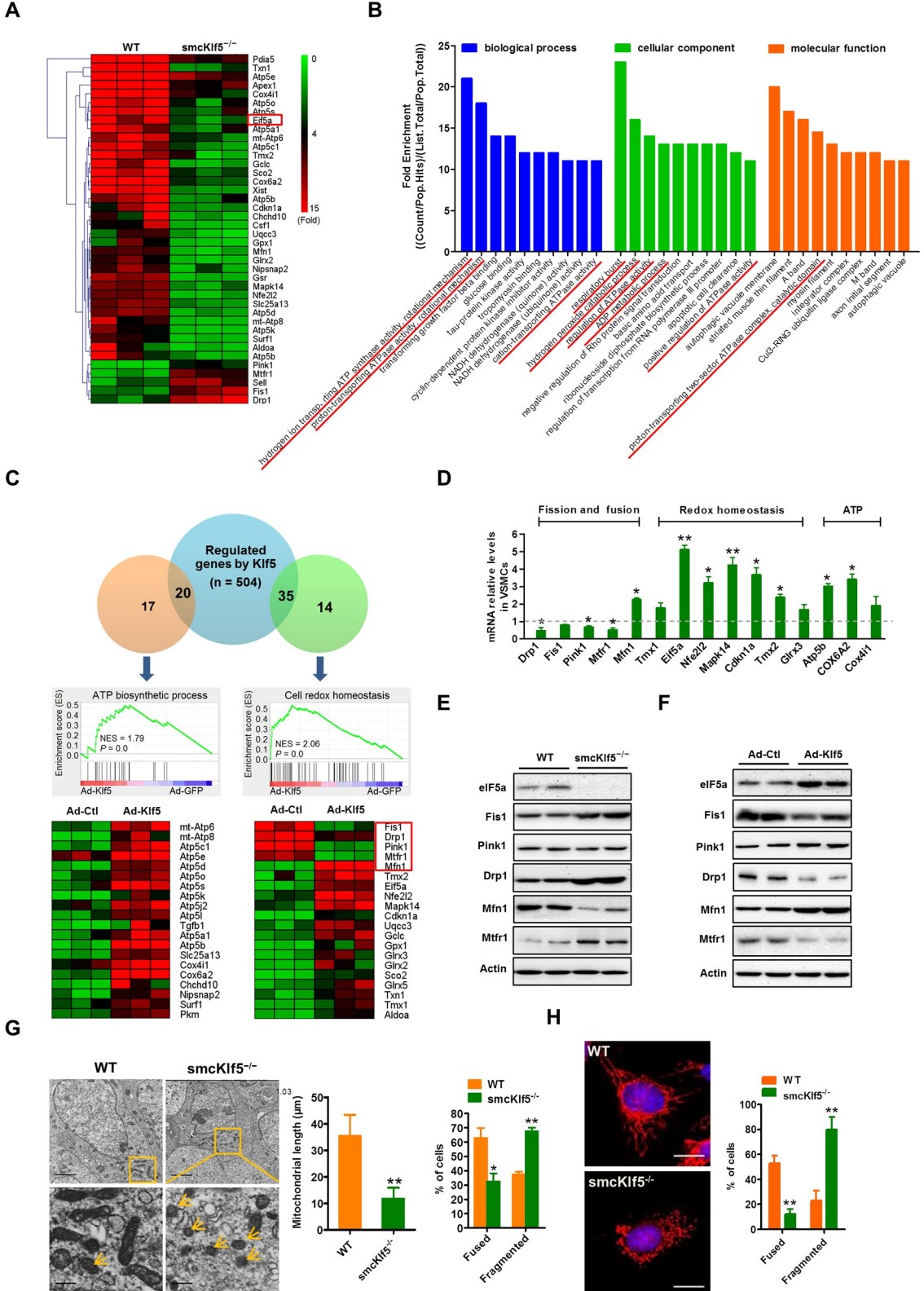

**Fig 4. Klf5 deficiency in mouse VSMCs leads to mitochondrial fission.** (A) Hierarchical clustering of the top 40 differentially expressed genes identified by microarray analysis in Ang II–injured aortas of WT and smcKlf5$^{-/-}$ mice ($n = 3$ for each group). Red color indicates high relative expression and green color indicates low relative expression. (B) The GO analysis for classification of the

differentially expressed genes based on the biological process (blue), cellular component (green), and molecular function (red). (C) Mouse VSMCs were infected with Ad-Ctl or Ad-Klf5 for 36 hours and then were analyzed by high-throughput mRNA sequencing. GSEA was used to identify the differential gene expression profiles between Klf5-overexpressing and control cells. (D) qRT-PCR validation of differential expression of a subset of genes, including the motochondial fission and fusion–, redox homeostasis–, and ATP biosynthesis–related genes. (E) Representative western blot of eIF5a, Fis1, Pink1, Drp1, Mfn1, and Mtfr1 in the aortas of WT and smcKlf5$^{−/−}$ mice. (F) Representative western blot of eIF5a, Fis1, Pink1, Drp1, Mfn1, and Mtfr1 in Ad-Ctl–and Ad-Klf5–infected mouse VSMCs. (G) Mitochondrial morphology in VSMCs from WT and Klf5$^{−/−}$ mice was detected by transmission electron microscopy. Statistical data of the mitochondrial length and percentage of cells containing fused and fragmented mitochondria are shown in histogram at the right side. Data are expressed as mean ± SEM. $n = 3$ per group, $^{*}P < 0.05$ and $^{**}P < 0.01$ versus WT. (H) MitoTracker Red-stained mitochondria in WT and Klf5$^{−/−}$ VSMCs. The nucleus was stained with DAPI. Right: the percentage of cells containing fused and fragmented mitochondria was quantified from more than 100 cells. Scale bars = 10 μm. Data represent mean ± SEM, $^{**}P < 0.01$ versus WT. For numerical raw data, please see S1 Data. For raw immunoblots, please see S1 Blots. Ad-Ctl, adenoviruses encoding control; Ad-Klf5, adenoviruses encoding Klf5; Ang II, angiotensin II; Drp1, dynamin-related protein 1; eIF5a, eukaryotic translation initiation factor 5a; Fis1, fission mitochondrial 1; GO, gene ontology; GSEA, Gene Set Enrichment Analysis; Klf5, Krüppel-like factor 5; Mfn1, mitofusin 1; Mtfr1, mitochondrial fission regulator 1; NADH, nicotinamide adenine dinucleotide phosphate; Pink1, PTEN-induced kinase 1; qRT-PCR, quantitative real-time PCR; VSMC, vascular smooth muscle cell; WT, wild-type.

subunit 6A isoform 2 (Cox6a2) were consistent with Klf5 (S9A and S9B Fig), but luciferase activity assay showed that these two genes were not regulated by Klf5 (S9C and S9D Fig). Because eIF5a is known to have a mitochondrial targeting property [32] and its expression was dramatically decreased in the aortic VSMCs of smcKlf5$^{−/−}$ mice (Fig 4A and 4E), we sought to know the effect of SMC-specific knockout of Klf5 on mitochondrial morphology. Thus, electron microscopy was used to examine the morphology of mitochondria. The results showed that the majority of mitochondria exist as rod-like shape (fused mitochondria) in the aortic VSMCs of WT mice. In contrast, in Klf5-deficient VSMCs, decreased mitochondrial length (fissed mitochondria) was observed, and the number of cells containing fragmented mitochondria was much higher in Klf5-deficient VSMCs than in WT cells (Fig 4G). Also, mitochondrial morphology was visualized in WT and Klf5-deficient VSMCs by immunofluorescent staining of mitotracker red. Similarly, the number of cells containing fused mitochondria was reduced, whereas fragmented mitochondria were increased in Klf5-deficient VSMCs (Fig 4H).

Simultaneously, the ATP content, mitochondrial membrane potential (ΔΨm) and the activity of mitochondrial respiratory chain complexes had a significant decrease in Klf5-deficient VSMCs compared with the WT cells, whereas adenovirus-mediated overexpression of Klf5 in Klf5-deficient VSMCs could rescue them to a large extent (S10A–S10E Fig). In addition, to identify whether impairment of mitochondrial functions is a cause or consequence of oxidative stress and eventually a major driver for AAA development, we detected the mitochondrial functions of VSMCs treated with Ang II for different times and assessed the markers for mitochondrial number (mtDNA) and replication (mitochondrial transcription factor A [mtTFA], peroxisome proliferative activated receptor, gamma, coactivator 1 alpha [PGC1α]). As shown in S10F–S10I Fig, Ang II treatment attenuated mitochondrial complex I, II, and IV activities and ATP levels in a time-dependent manner. Correspondingly, Ang II also decreased time-dependently the protein expression of PGC-1α and mtTFA, as well as the number of mtDNA. We speculated that reciprocal causation between mitochondrial function impairment and oxidative stress co-operatively drives AAA development and progression.

## Klf5 activates the transcription of eIF5a gene through direct binding to its promoter

Because eIF5a was down-regulated in the aortic VSMCs of smcKlf5$^{−/−}$ mice (Fig 4D and 4E), we sought to determine whether there exists a causal relationship between eIF5a down-regulation and Klf5 deficiency, as well as between eIF5a down-regulation and time course of Ang II

stimulation. First, we examined the effect of Ang II on the expression of eIF5a. In parallel with alterations in Klf5 expression (**Fig 3C**), the expression of eIF5a was markedly increased 1 day after Ang II treatment, but significantly decreased in Ang II–treated mouse VSMCs for 3 and 5 days (**Fig 5A**). A similar result was obtained by qRT-PCR (**Fig 5B**). Furthermore, we infected VSMCs with Ad-Klf5 or Ad-shKlf5 to overexpress or knock down Klf5. As expected, Klf5 overexpression increased, whereas Klf5 knockdown decreased eIF5a expression at both transcription and translation levels compared with their corresponding controls (**Fig 5C and 5D**). These results suggest that Klf5 plays an important role in the regulation of eIF5a expression.

In further experiments, we investigated whether Klf5 regulates the expression of eIF5a by direct binding to its promoter. Using the JASPAR CORE database, we performed a Klf5-binding motif analysis on the eIF5a promoter and identified three typical Klf5-binding sites in the −1,500 to +1 bp of the 5′ upstream promoter of eIF5a gene (**Fig 5E**). Subsequently, a series of 5′-deletion mutants of the eIF5a promoter were constructed and tested by the luciferase activity assay. The results showed that enforced Klf5 expression markedly increased the activity of the eIF5a full-length promoter. Deletion of −674 to −287 bp or −287 to −89 bp of the eIF5a promoter region significantly decreased the activation by Klf5, indicating that Klf5-binding sites 1 and 2 between −300 bp and −152 bp are critical for the Klf5-mediated transcriptional activation of the eIF5a promoter (**Fig 5F**). To further validate these results, we carried out ChIP and oligonucleotide pull-down assays. The results showed that although the slight binding of Klf5 to the sites 1 and 2 of the eIF5a promoter could be detected under basal conditions, Ang II treatment obviously increased the recruitment of Klf5 to these two sites (**Fig 5G and 5H**). Mutation of these two sites abolished Klf5 binding (**Fig 5H**), indicating that the binding of Klf5 is specific. These results suggest that Klf5 activates the transcription of eIF5a gene by direct binding to its promoter and Klf5 down-regulation is responsible for decreased expression of eIF5a.

### eIF5a preserves mitochondrial integrity through interacting with Mfn1, down-regulation of eIF5a elicited by Klf5 deficiency results in mitochondrial fission

Because alterations of mitochondrial dynamics, which is controlled by fission and fusion, induce the mtROS generation [33], and eIF5a is localized not only to the nucleus but also to the mitochondria [24] and might be involved in the regulation of redox homeostasis [25], we wanted to determine whether eIF5a down-regulation induced by Klf5 deficiency affects mitochondrial dynamics. Thus, the interaction of eIF5a with mitochondrial dynamics–related proteins, such as Mfn1, Drp1, Fis1, and Mtfr1, was examined in VSMCs stimulated chronically by Ang II. The co-immunoprecipitation assay confirmed that the interaction of eIF5a with fusion protein Mfn1 was dramatically enhanced at 1 day after Ang II stimulation, but then markedly reduced at 5 days (**Fig 6A**). Also, an increased interaction between eIF5a and Mfn1 was detected 1 day after Ang II stimulation by in situ proximity ligation assay (**Fig 6B**). These changes are very similar to those of Klf5 and eIF5a expression following chronic Ang II stimulation. However, no significant interaction between eIF5a and fission factors Drp1, Fis1, or Mtfr1 was observed by co-immunoprecipitation assay (**S11 Fig**). To provide additional confirmation that eIF5a interacts with Mfn1, co-immunofluorescence staining was performed using anti-eIF5a, anti-Mfn1, and mitotracker in eIF5a-overexpressed or knocked down VSMCs. Although eIF5a and Mfn1 co-localization in the mitochondria was detectable in empty vector–transduced VSMCs, eIF5a overexpression obviously increased their co-localization and facilitated the formation of network-like mitochondria (fused mitochondria), whereas eIF5a knockdown had the opposite effects (**Fig 6C, S12A Fig**). Moreover, eIF5a overexpression obviously

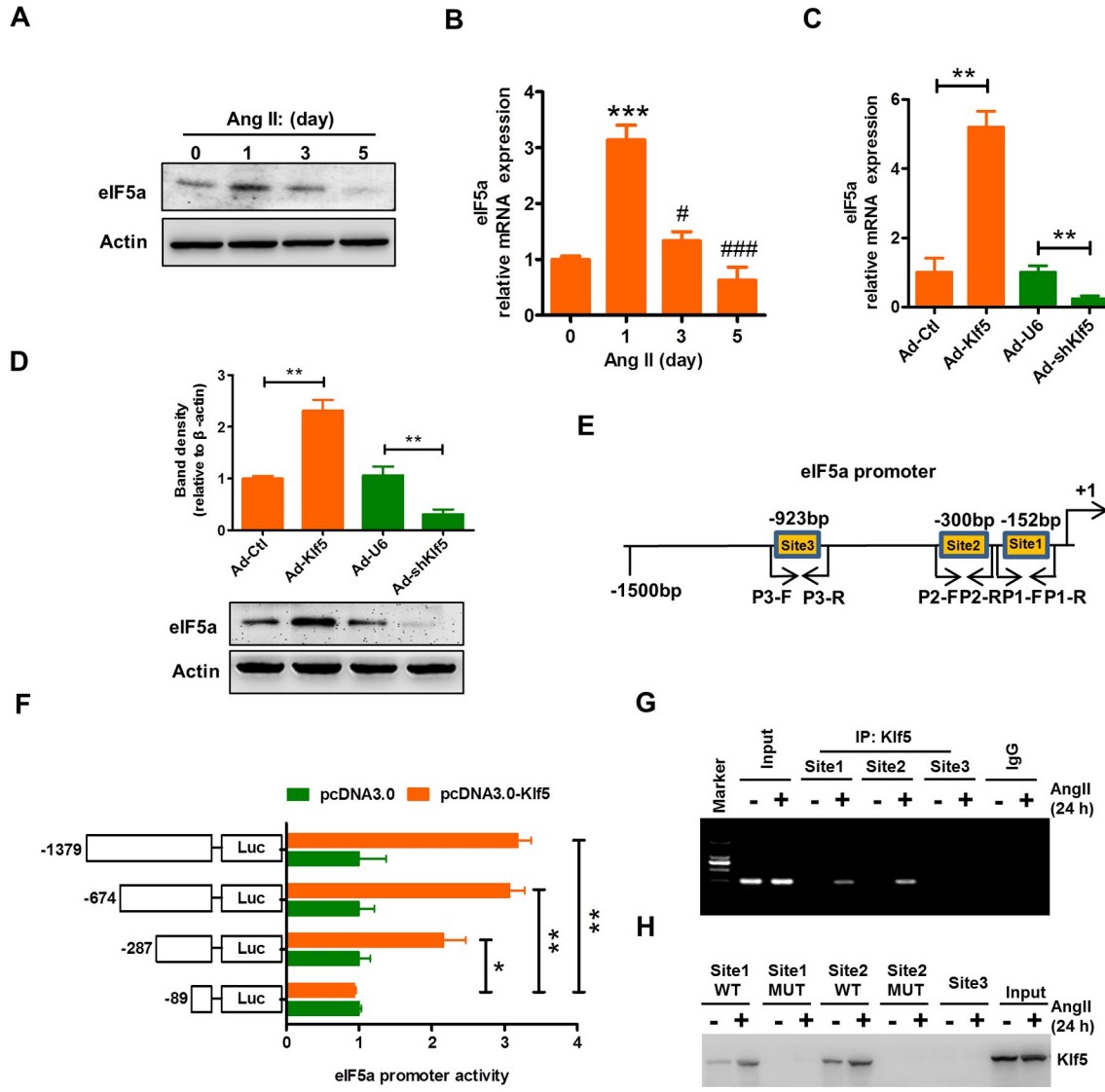

**Fig 5. Klf5 activates the transcription of the eIF5a gene through direct binding to its promoter.** (A,B) Mouse VSMCs were stimulated with Ang II (100 nmol/L) for the indicated times ($n$ = 3 in each group). The expression of eIF5a was analyzed by western blotting (A) and qRT-PCR (B). β-actin was used as a loading control. $^{***}P < 0.001$ versus 0 day, $^{#}P < 0.05$ and $^{##}P < 0.01$ versus 1 day. (C,D) Mouse VSMCs were infected with Ad-Ctl, Ad-Klf5, Ad-U6, and Ad-shKlf5 for 36 hours. qRT-PCR (C) and Western blotting (D) detected mRNA and protein expression of eIF5a. β-actin was used as a loading control. $^{**}P < 0.01$ versus their corresponding controls. (E) A schematic map of the −1,500 to +1 bp region of the human eIF5a promoter, showing the position of 3 Klf5-binding sites and primers used for amplification. (F) Cells (293A) were transfected with the reporter directed by the eIF5a promoter containing different 5′-deletion fragments, and luciferase activity was measured. Data represent the relative eIF5a promoter activity normalized to pRL-TK activity. $^{*}P < 0.05$ and $^{**}P < 0.01$ versus the reporter containing the −89 to +1 bp region. (G) Mouse VSMCs were incubated with or without Ang II (100 nmol/L) for 24 hours. ChIP assay was then performed with antibody against Klf5. Nonimmune IgG was used as negative control. Immunoprecipitated DNA was amplified by PCR using the primers indicated as in (E). (H) An oligo pull-down assay was done with Ang II–treated mouse VSMC lysates and biotinylated double-stranded oligonucleotide containing the Klf5-binding sites 1, 2, and 3 (WT and MUT) as probes. The DNA-bound protein was detected by western blotting with anti-Klf5 antibody. For numerical raw data, please see S1 Data. For raw immunoblots, please see S1 Blots. Ad-Ctl, adenoviruses encoding control; Ad-Klf5, adenoviruses encoding Klf5; Ad-shKlf5, adenoviruses encoding small hairpin Klf5; Ad-U6, adenoviruses encoding U6; Ang II, angiotensin II; ChIP, chromatin immunoprecipitation; eIF5a, eukaryotic translation initiation factor 5a; IgG, immunoglobulin G; IP, immunoprecipitation; Klf5, Krüppel-like factor 5; Luc, luciferase; MUT, mutation; pRL-TK, thymidine kinase promoter-Renilla luciferase reporter plasmid; qRT-PCR, quantitative real-time PCR; VSMC, vascular smooth muscle cell; WT, wild-type.

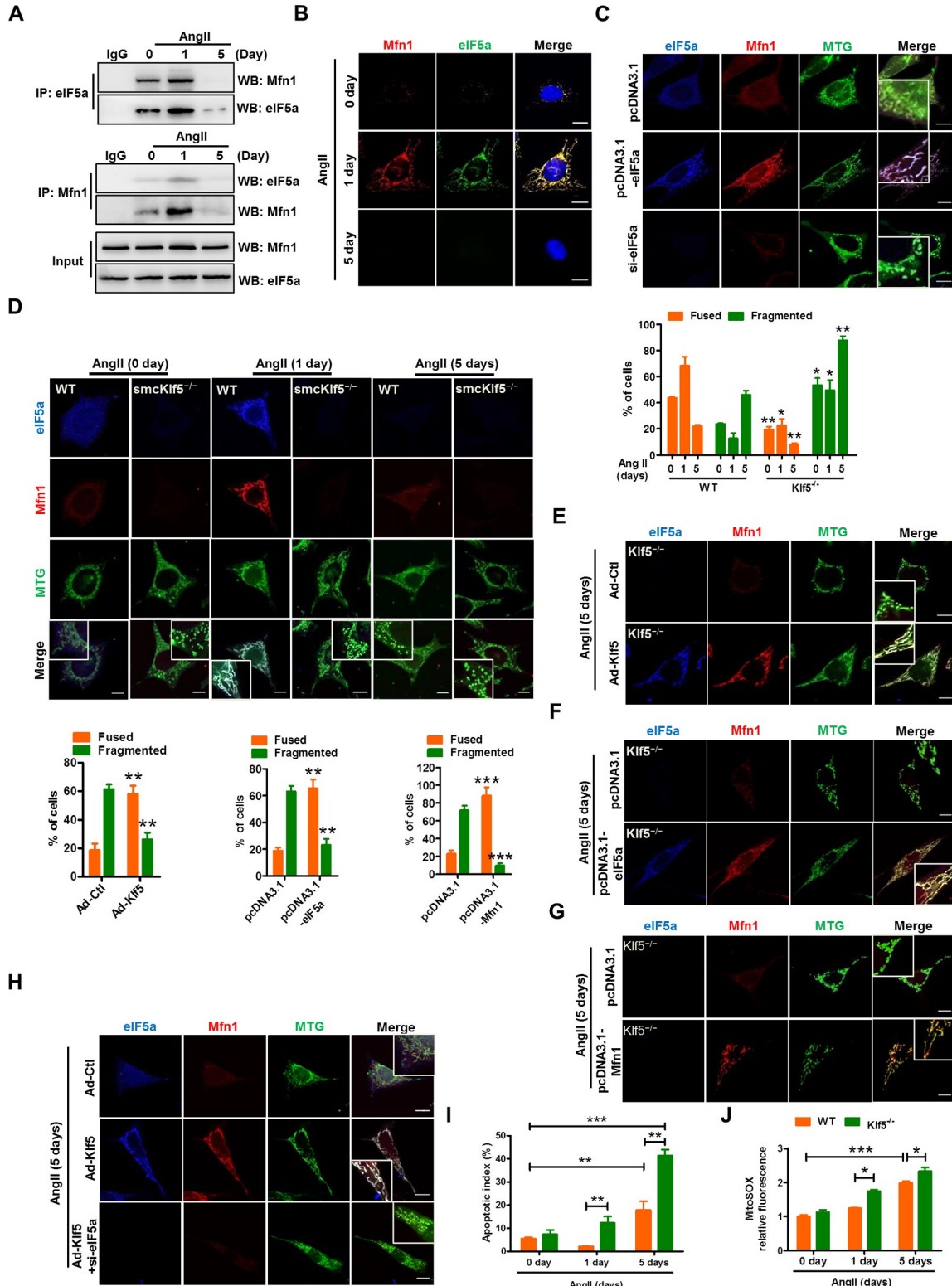

**Fig 6. eIF5a interacts with Mfn1 and preserves mitochondrial integrity.** (A) Reciprocal co-immunoprecipitation between eIF5a and Mfn1 in mouse VSMCs treated with Ang II (100 nmol/L) for the indicated times. IgG was used as a negative control. (B) Duolink detection system (in situ PLA) was used to detect eIF5a and Mfn1 interaction in situ in mouse VSMCs treated as in (A). Scale bars = 10 μm. (C) Mouse VSMCs were transfected with pcDNA3.1, pcDNA3.1-eIF5a, or si-eIF5a for 36 hours. Immunofluorescence staining of eIF5a and Mfn1 as well as MitoTracker Green (MTG) mitochondria staining were performed and observed by confocal

microscopy. Scale bar = 10 μm. (D) Mouse WT and Klf5$^{-/-}$ VSMCs were treated as in (A). Immunofluorescence staining of eIF5a (blue), Mfn1 (red), and MTG (green) was used to visualize mitochondrial morphology. Scale bars = 10 μm. Percentage of cells containing fused and fragmented mitochondria is shown at the right side. Data represent mean ± SEM of 3 independent experiments in which 300 cells were analyzed. $^*P < 0.05$ and $^{**}P < 0.01$ versus WT. (E-G) Mouse Klf5$^{-/-}$ VSMCs were infected with Ad-Ctl and Ad-Klf5 (E) or transfected with pcDNA3.1 and pcDNA3.1-eIF5a (F) or pcDNA3.1-Mfn1 (G) for 36 hours. Mitochondrial morphology was visualized as described above. Scale bars = 10 μm. Left: Percentage of cells containing fused and fragmented mitochondria is shown as histograms. The data represent mean ± SEM of 3 independent experiments in which 300 cells were analyzed. $^{**}P < 0.01$, $^{***}P < 0.001$ versus their corresponding control. (H) Mouse VSMCs were infected with Ad-Ctl, Ad-Klf5, or Ad-Klf5 plus si-eIF5a for 36 hours and then treated with Ang II for 5 days. Mitochondrial morphology was visualized as described above. Scale bars = 10 μm. (I,J) Mouse WT and Klf5$^{-/-}$ VSMCs were treated with Ang II for 0, 1, 5 days, TUNEL and mitoSOX staining was performed, and then TUNEL-positive (I) and mitoSOX-positive (J) cells were quantified from three different fields. Data represent mean ± SEM, $^*P < 0.05$, $^{**}P < 0.01$, $^{***}P < 0.001$ versus WT or 0 day. For numerical raw data, please see S1 Data. For raw immunoblots, please see S1 Blots. Ad-Ctl, adenoviruses encoding control; Ad-Klf5, adenoviruses encoding Klf5; Ang II, angiotensin II; eIF5a, eukaryotic translation initiation factor 5a; IgG, immunoglobulin G; Klf5, Krüppel-like factor 5; Mfn1, mitofusin 1; MTG, MitoTracker Green; PLA, proximity ligation assay; si-eIF5a, short interfering RNA targeting eIF5a; VSMC, vascular smooth muscle cell; WB, western blot; WT, wild-type.

increased the expression of Mfn1 (**S12B and S12C Fig**). Reversely, the expression of mitochondrial fission factors Fis1 and Mff was significantly down-regulated by eIF5a overexpression, without affecting the expression of other proteins detected in the present study, as evidenced by western blot analysis (**S12C Fig**). These data suggest that eIF5a and Mfn1 are co-localized to the mitochondria and might play a functional role in the regulation of mitochondrial dynamics.

Furthermore, using cell immunofluorescent staining of mitotracker, eIF5a and Mfn1, we investigated the effects of chronic Ang II stimulation on mitochondrial dynamics in WT and Klf5-deficient VSMCs. When stimulated with Ang II for 1 day, the majority of mitochondria in WT VSMCs formed typical network-like structures. In contrast, when WT and Klf5-deficient VSMCs were treated with Ang II for 5 days, the majority of mitochondria displayed fragmented structures (fissed mitochondria) (**Fig 6D**). Correspondingly, the number of cells containing fused mitochondria was significantly increased in WT VSMCs treated with Ang II for 1 day. Conversely, upon exposure to Ang II for 5 days, the number of cells containing fragmented mitochondria was markedly elevated not only in WT cells but also in Klf5-deficient VSMCs (**Fig 6D**). These findings indicate that both loss of Klf5 and chronic Ang II stimulation lead to mitochondrial fission in VSMCs.

To corroborate these findings, we performed rescue experiments in Klf5-deficient VSMCs. The results indicated that not only Klf5- but also eIF5a- or Mfn1-enforced expression could enable the fragmented mitochondria to fuse, forming a network-like structure (**Fig 6E–6G**). And eIF5a-enforced expression increased Mfn1 expression, whereas Mfn1-enforced expression did not affect eIF5a expression (**S13A and S13B Fig**). Correspondingly, we overexpressed Klf5, elF5a, or Mfn1 by transfecting Klf5$^{-/-}$ VSMCs with expressing plasmids encoding Klf5, elF5a, or Mfn1, respectively, and then treated cells with Ang II for 24 hours. The results showed that enforced expression of Klf5, elF5a, or Mfn1 in Klf5$^{-/-}$ VSMCs greatly reduced Ang II–stimulated mitochondrial fission compared with empty vector–transfected Klf5$^{-/-}$ VSMCs (**S14 Fig**). Additionally, eIF5a knockdown could abrogate Klf5 overexpression–induced mitochondrial fusion (**Fig 6H**), and overexpression of Klf5 or eIF5a partly rescued Ang II–induced decrease in ATP production (**S15 Fig**). Together, these data indicate that eIF5a or Mfn1 resides downstream of Klf5 and is involved in the regulation of mitochondrial dynamics. Additionally, we also found that knockdown of Mfn1 enhanced mitochondrial fission and abolished eIF5a enforced expression-promoted mitochondrial fusion (**S16 Fig**). Moreover, increased mitochondrial fission was also accompanied by the increased apoptosis and the elevated mtROS levels in Ang II–stimulated VSMCs for 5 days (**Fig 6I and 6J**).

## Inhibition of the mitochondrial fission decreases mtROS production

To further clarify the relationship between mitochondrial fission and mtROS generation, we utilized two approaches to inhibit mitochondrial fission and then observed mtROS generation. First, compared with short interfering RNA control (si-Ctl)–transfected VSMCs, knockdown of Drp1, a key element in mitochondrial fission, by short interfering RNA targeting dynamin related protein 1 (si-Drp1) could significantly suppress mitochondrial fission in VSMCs stimulated with Ang II for 5 days (**Fig 7A**). Furthermore, we confirmed that pharmacological inhibition of Drp1 with Mitochondrial division inhibitor 1 (Mdivi-1), a small molecule inhibitor of Drp1 [34,35], also markedly inhibited mitochondrial fission induced by chronic Ang II stimulation (**Fig 7C**). Correspondingly, the mtROS production was also reduced obviously in si-Drp1–transfected (**Fig 7B**) or Mdivi-1–treated VSMCs (**Fig 7D**).

It has been known that NAPDH oxidase (NOX)1 and NOX4 are key enzymes responsible for ROS generation in VSMCs [36]. To understand the source of ROS and to determine the association of Ang II signaling, Klf5, and eIF5α with NOX expression, mouse VSMCs were transfected with expression vectors for Klf5 and eIF5α or small interfering RNA (siRNA) against Klf5 and eIF5α, respectively, and then treated with Ang II for 0, 1, 3, and 5 days, and the expression of NOX1 and NOX4 was detected by RT-qPCR. As shown in **S17A–S17D Fig**, Ang II treatment increased NOX1 expression, peaked at 24 hours after treatment, and gradually declined thereafter. Overexpression of Klf5 or eIF5a significantly decreased, whereas their knockdown increased, NOX1 expression. However, single treatment with Ang II did not significantly affect NOX4 expression. We further confirmed these observations at the protein level by western blot (**S17E and S17F Fig**). These findings suggest that Ang II promotes ROS generation at least in part via up-regulating NOX1 expression, whereas Klf5 or eIF5a decreases Ang II–induced ROS formation through suppressing NOX1 expression.

To further understand the extent of the involvement of oxidative stress in AAA development, mouse VSMCs were pretreated with N-acetyl-L-cysteine (NAC; an ROS scavenger), followed by treatment with Ang II for 1 or 5 days. The results showed that NAC pretreatment significantly attenuated mtROS levels in Ang II–stimulated VSMCs for 1 or 5 days (**S18A Fig**). Moreover, pretreatment with NAC significantly reduced Ang II–induced mitochondrial fission in Ang II–treated VSMCs for 1 day relative to those untreated with NAC, but NAC had little inhibitory effect on mitochondrial fission in Ang II–treated cells for 5 days (**S18B Fig**). These results suggest that antioxidative treatment with the ROS scavenger may partially eliminate excessive ROS in the early stage of AAA, preventing ROS-induced mitochondrial fission.

To assess the effect of the inhibition of mitochondrial fission by Mdivi-1 on AAA formation, Mdivi-1 (25 mg/kg every other day) was administered by intraperitoneal injection to ApoE$^{-/-}$ and ApoE$^{-/-}$ smcKlf5$^{-/-}$ mice, followed by chronic Ang II infusion for 28 days. The results showed that the aortic external diameter of ApoE$^{-/-}$ smcKlf5$^{-/-}$ mice had a >20% dilation relative to ApoE$^{-/-}$ mice. Importantly, administration of Mdivi-1 significantly decreased Ang II–induced aortic dilation, regardless of Klf5 deletion in VSMCs (**Fig 7E and 7F**). Furthermore, transmission electron microscopy was used to observe mitochondrial morphology and revealed that administration of Mdivi-1 to ApoE$^{-/-}$ or ApoE$^{-/-}$ smcKlf5$^{-/-}$ mice significantly increased the mitochondrial length relative to their corresponding controls (**Fig 7G and 7H**). Also, we demonstrated the decreased mitochondrial fission was accompanied by the reduced mtROS production (**Fig 7I**) and cell apoptosis (**S19 Fig, Fig 7J**). Collectively, these results suggest that genetic and pharmacological inhibition of the mitochondrial fission can decrease mtROS production and AAA formation.

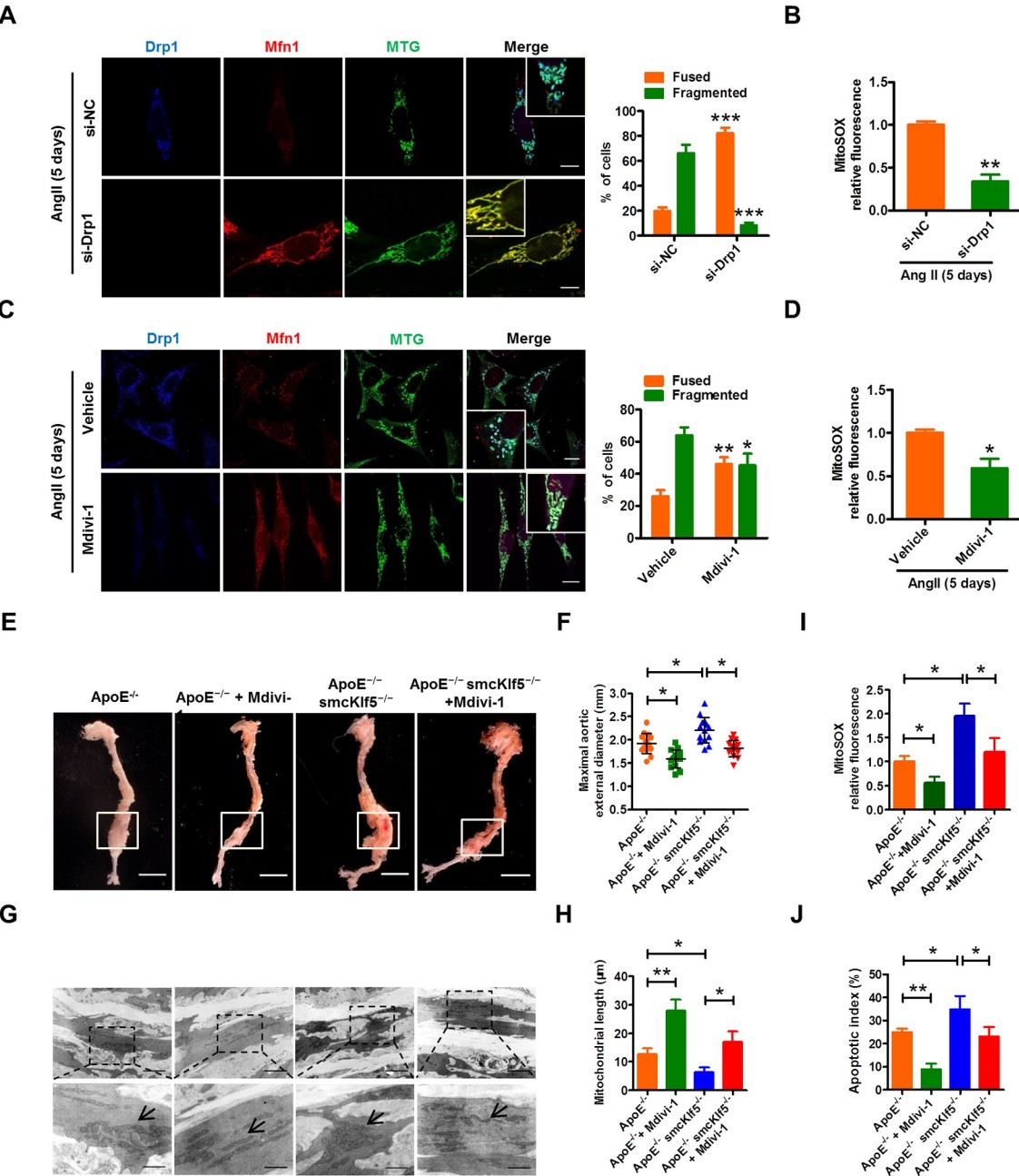

**Fig 7. Inhibition of the mitochondrial fission decreases mtROS production.** (A, C) Mouse VSMCs were transfected with si-NC and si-Drp1 (A) or treated with vehicle and Mdivi-1 (inhibitor of Drp1) (C) for 24 hours, and then stimulated with Ang II (100 nmol/L) for 5 days. Mitochondrial morphology was visualized by staining with MTG, anti-Drp1, and anti-Mfn1 antibodies. Scale bars = 10 μm. Right: Percentage of cells containing fused and fragmented mitochondria is shown as histograms. The data represent mean ± SEM of 3 independent experiments in which 300 cells were analyzed. *$P < 0.05$, **$P < 0.01$, and ***$P < 0.001$ versus their corresponding control. (B,D) Quantitative analysis of intracellular mtROS in VSMCs treated with si-Drp1 (B) or Mdivi-1(D). Data are expressed as mean ± SEM. *$P < 0.05$ and **$P < 0.01$ versus their corresponding control. (E) Representative photographs of AAAs induced by Ang II infusion for 28 days in ApoE$^{-/-}$ and ApoE$^{-/-}$ smcKlf5$^{-/-}$ mice treated with or without Mdivi-1 for 4 weeks. Scale bars = 2.5 mm. (F) Statistical analysis of the maximal aortic external diameter. Data represent the mean ± SEM. *$P < 0.05$ versus ApoE$^{-/-}$ mice or ApoE$^{-/-}$ smcKlf5$^{-/-}$ mice. $n = 8$ for each group. (G) Transmission electron microscopy reveals mitochondrial morphology in Ang II–injured aortas of ApoE$^{-/-}$ or ApoE$^{-/-}$ smcKlf5$^{-/-}$ mice treated as in (E). (H) Statistical data of the quantified mitochondrial length in each group. Data are presented as the means ± SEM. *$P < 0.05$, **$P < 0.01$ versus ApoE$^{-/-}$ mice or ApoE$^{-/-}$ smcKlf5$^{-/-}$ mice. $n = 3$ in each group. (I,J) Graphical data show mitoSOX fluorescence of VSMCs (I) and the percentage of apoptotic cells (J). All data are presented as the means ± SEM. *$P < 0.05$ and **$P < 0.01$ versus ApoE$^{-/-}$ mice or ApoE$^{-/-}$ smcKlf5$^{-/-}$ mice. For numerical raw data, please see S1 Data. AAA, abdominal aortic aneurysm; Ang II, angiotensin II; Drp1, dynamin-related protein 1;

Mdivi-1, mitochondrial division inhibitor 1; Mfn1, mitofusin 1; MTG, MitoTracker Green; mtROS, mitochondrial ROS; si-Drp1, short interfering RNA targeting Drp1; si-NC, short interfering RNA control; VSMC, vascular smooth muscle cell.

### eIF5a expression is up-regulated in unruptured AAAs and decreased in ruptured AAAs

To corroborate whether abnormal expression of eIF5a is involved in the development and progression of human AAAs, we examined the expression level of eIF5a in the normal aorta, unruptured AAA, and ruptured AAA. The results showed that eIF5a expression was obviously higher in unruptured AAA than in the normal abdominal aorta, but its expression was significantly decreased in ruptured AAA compared with the unruptured AAA, as evidenced by qRT-PCR and immunofluorescence staining (**Fig 8A and 8B**). Moreover, eIF5a was localized to VSMCs in unruptured AAA tissues (**Fig 8B**). Statistical analysis revealed a strong correlation between the expression level of eIF5a and AAA size (**Fig 8C**). These results indicate that eIF5a down-regulation induced by Klf5 deficiency in the aortic VSMCs is correlated with the progression and rupture of human aortic aneurysm.

## Discussion

The major findings of the present study were as follows: (1) Klf5 reduction in VSMCs is related to the rupture of human AAA. (2) Loss of Klf5 in VSMCs exacerbates VSMC senescence by enhancing mitochondrial fission and ROS production. (3) Klf5 activates the transcription of

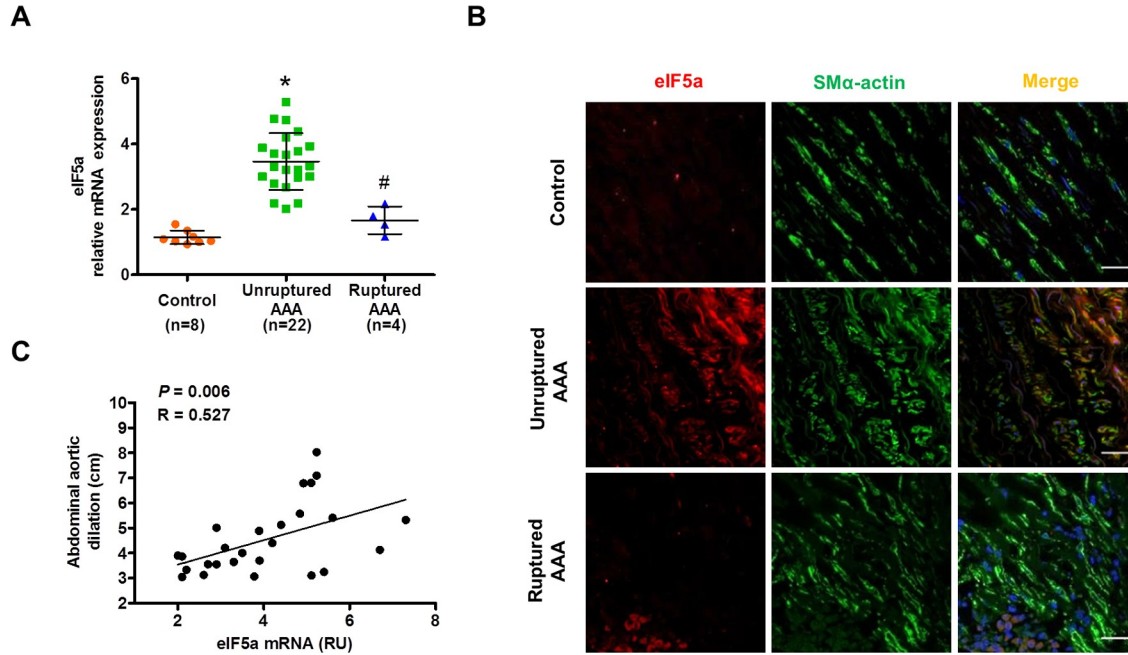

**Fig 8. eIF5a is up-regulated in unruptured AAAs and decreased in ruptured AAAs.** (A) qRT-PCR detected eIF5a mRNA in human unruptured ($n = 22$), ruptured ($n = 4$) AAA, and control aortic tissues (control, $n = 8$), $^*P < 0.05$ versus control, $^{\#}P < 0.05$ versus unruptured AAA. (B) Representative confocal fluorescence images of SM α-actin (green), eIF5a (red), and DAPI (blue) on human control, unruptured, and ruptured AAA sections. Scale bars = 10 μm. (C) Correlation analysis between eIF5a mRNA level and AAA size. Spearman's correlation coefficients were used to test the strength of a linear relationship between eIF5a mRNA and AAA size. Spearman's correlation coefficient and $P$ value are shown in graph. For numerical raw data, please see S1 Data. AAA, abdominal aortic aneurysm; eIF5a, eukaryotic translation initiation factor 5a; qRT-PCR, quantitative real-time PCR; RU, relative unit; SM, smooth muscle.

eIF5a gene through direct binding to the promoter of eIF5a, which in turn preserves mitochondrial integrity by interacting with Mfn1. eIF5a down-regulation elicited by Klf5 deficiency results in mitochondrial fission. (4) Inhibition of mitochondrial fission by genetic and pharmacological approaches decreases ROS production and VSMC senescence. (5) eIF5a expression is decreased in ruptured human AAA tissues.

Although Ang II is a key regulator of blood pressure and VSMC proliferation, chronic Ang II stimulation also induces senescence of vascular cells [37,38]. Moreover, Ang II is a potent mediator of oxidative stress and oxidant signaling leading to vascular premature senescence [28]. In this study, we showed that short-term exposure (1 day) of the cultured VSMCs to Ang II markedly increased Klf5 expression, while chronic Ang II stimulation (5 days) dramatically reduced Klf5 expression levels, accompanied simultaneously by an increase in the number of senescent VSMCs or a decrease in proliferating cells. In parallel with VSMC senescence, Klf5 knockdown, or overexpression in VSMCs markedly enhanced or suppressed, respectively, Ang II–induced total ROS and the mtROS generation. It has been demonstrated that oxidative stress not only results in DNA damage but also makes cells exit the cell cycle by inducing expression of negative cell cycle control genes p57kip2 and p21 [7,11]. Our results along with previous studies indicate that excessive ROS production induced by Klf5 deficiency leads to the lower proliferation capacity and the appearance of premature senescence in VSMCs. These are characteristic features of the age-related vascular disorders, such as atherosclerosis, AAA, hypertension, and diabetes.

Mitochondria exist in a dynamic equilibrium between fragmented and fused states, and mitochondrial dynamics is controlled by fission and fusion [17,33]. Increased mitochondrial fission and/or attenuated fusion lead to mitochondrial fragmentation and disrupt cellular physiological function [38]. Previous studies indicate that the increased mitochondrial fission leads to mtROS generation and elevated apoptosis in endothelial cells (ECs) of diabetic patients [31,39–41]. Mitochondrial fragmentation, increased expression of the fission factor, and reduced fusion protein levels were demonstrated in ECs of type 1 diabetic mice [42]. In this study, we discovered that knockdown of Klf5 enhanced, while overexpression of Klf5 reduced mitochondrial fission induced by chronic Ang II stimulation. To further explore the relationship between mitochondrial fission triggered by Klf5 down-regulation and ROS accumulation, we used microarray analysis to identify the mitochondrial dynamics–related genes regulated by Klf5. As expected, Drp1, Fis1, and Mtfr1 were significantly up-regulated, whereas Mfn1 was decreased in Ang II–injured aortas of smcKlf5$^{-/-}$ mice as well as in the aortas of older WT and smcKlf5$^{-/-}$ mice, but the opposite results were obtained in Klf5-overexpressed VSMCs. Recent reports show that transfection of an miRNA-142a-5p mimic into skeletal muscle cells (C2C12) down-regulates Mfn1 expression, leading to extensive mitochondrial fragmentation, depolarization of Δψm, and accumulation of ROS, with a concomitant mitophagy activation and apoptosis. Importantly, these effects can be attenuated by overexpression of Mfn1. [43,44]. Consistently, our results showed that chronic Ang II stimulation–induced Klf5 down-regulation contributed to low expression of eIF5a in VSMCs, which in turn decreased the expression of Mfn1 and thus attenuated the interaction of eIF5a with Mfn1, subsequently leading to excessive accumulation of ROS and VSMC senescence. Therefore, impaired mitochondrial function might be caused by the aberrant expression of mitochondria dynamics–related genes. These are consistent with previous studies demonstrating that oxidative and other stresses frequently induce fission rather than fusion of mitochondria [45]. Accordingly, dysregulation of mitochondrial dynamics alters mitochondrial morphology and may affect mitochondrial function, accompanied by a decrease in ATP biosynthesis and an increase in mtROS generation.

Besides mitochondrial dynamics–related genes, eIF5a was highly expressed in Klf5-overexpressed VSMCs. Although eIF5a was originally believed to be a translation initiation factor

that stimulates the initiation phase of protein synthesis by transient association with ribosomes [46], more recent studies show that eIF5a has numerous other functions. For example, hypusinated eIF5a modulates mitochondrial respiration and macrophage activation via promoting the efficient expression of a subset of mitochondrial proteins involved in the tricarboxylic acid cycle and oxidative phosphorylation [47]. eIF5a is also required for efficient translation of mRNAs encoding proteins with poly(Pro) tracts (such as PPP or PPG) by preventing ribosomes from stalling at such sequences [32,48]. Tyrosine sulfated-eIF5a is secreted from cardiac myocytes in response to hypoxia/reoxygenation and mediates oxidative stress–induced apoptosis [49]. Inhibition of nuclear export protein exportin 1 (XPO1) causes eIF5a accumulation in the mitochondria and leads to the induction of apoptosis [24]. Reduced eIF5a function affects cell growth and autophagy via ATG3 protein synthesis [50,51]. Moreover, eIF5a was identified to be a significantly differentially expressed protein involved in the regulation of redox homeostasis [25]. In this study, we first demonstrated that Klf5 directly bound to the eIF5a promoter and activated its transcription. Both Klf5 down-regulation and chronic stimulation of VSMCs by Ang II dramatically attenuated the expression of eIF5a. Also, eIF5a expression was significantly reduced in aortic VSMCs of smcKlf5$^{-/-}$ mice. These data suggest that eIF5a down-regulation is responsible for Ang II–induced VSMC senescence, consistent with previous studies showing that a reduction of eIF5a content is associated with brain aging [52].

Importantly, using co-immunoprecipitation assay, in situ proximity ligation assay, and confocal immunofluorescence staining, we showed that eIF5a and Mfn1 were co-localized to the mitochondria. eIF5a overexpression obviously increased their co-localization and facilitated the formation of network-like mitochondria (fused mitochondria), whereas eIF5a knockdown had the opposite effects, accompanied simultaneously by a corresponding change in mitochondrial morphology. Mfn1 is a ubiquitous and well-conserved GTPase responsible for regulating mitochondrial dynamics and bioenergetics [53]. Mfn1-mediated mitochondrial fusion is necessary for mitochondrial morphology and cell health. In the present study, sustained Ang II stimulation resulted in eIF5a down-regulation and thus decreased Mfn1 and eIF5a interaction, which is critical to protect against accumulation of fragmented/dysfunctional mitochondria, subsequently leading to mitochondrial dysfunction and cell death associated with the pathophysiology of aneurysm. Together, these data suggest that eIF5a can be localized to the mitochondria and protects them from Klf5 deficiency–induced fragmentation. However, despite recent advances in our understanding of eIF5a functions, the excise mechanisms whereby mitochondrial integrity is maintained through interaction of eIF5a with Mfn1 need to be further elucidated.

Recent studies show that the expressing alteration of Drp1 and/or its mitochondrial translocation regulates mitochondrial fission and apoptosis [54,55], and that VSMC apoptosis could be targetable mechanisms implicated in the pathogenesis of AAA formation [56]. Our results show that loss of Klf5 in VSMCs resulted in Drp1 up-regulation and mitochondrial translocation, which are requisite events in Drp1-dependent mitochondrial fragmentation and cell apoptosis. To investigate the contribution of Drp1 to the pathophysiology of AAA, we delivered Mdivi-1, a selective inhibitor of Drp1, to mouse AAA model. Our results revealed that pharmacologic inhibition of mitochondrial fission factor Drp1 could significantly attenuate mitochondrial fission and the mtROS production induced by chronic Ang II stimulation in VSMCs. Administration of Mdivi-1 to ApoE$^{-/-}$ mice significantly decreased Ang II–induced AAA formation. These results are consistent with previous studies demonstrating that Mdivi-1 protects rat hippocampal neural stem cells against palmitate-induced oxidative stress and apoptosis by preserving mitochondrial integrity [57]. These findings suggest that the prevention or inhibition of excessive mitochondrial fission may have therapeutic value in age-related vascular diseases. Additionally, in this study, we demonstrated that Klf5-dependent regulation of

eIF5a/Mfn1 is required for mitochondrial dynamics and homeostasis processes, and that eIF5a reduction elicited by Klf5 down-regulation contributes to mitochondrial fission, which in turn leads to excessive ROS generation and thus enhances VSMC senescence. These findings suggest that antioxidative treatment with the ROS scavenger would be desirable for prevention and treatment of AAA in the early stage of the disease, which will be highlighted in the further research.

A major limitation of the present study is that we did not study the protective effect of the antioxidant NAC on Ang II–induced vascular injury. Moreover, in the cultured VSMCs, suppression of Ang II–induced oxidative stress by NAC only partially improved mitochondrial morphology. In addition, Ang II–induced production of inflammatory factors also plays an important role in the impairment of mitochondrial function and vascular senescence. Certainly, multiple mechanisms, such as oxidative stress, inflammatory response, and dysregulated mitochondrial dynamics, are involved in vascular senescence and AAA development and progression. Therefore, treatment of vascular senescence and AAA with antioxidants can be only supplemental to other interventions that should target more essential pathways for the development of the disease.

Collectively, our studies discover a previously unrecognized role of eIF5a in the regulation of mitochondrial dynamics. Targeting the Klf5-eIF5a/Mfn1 regulatory pathway provides a potential therapeutic strategy for age-related vascular disorders.

## Material and methods

### Ethics statement

All human tissue specimens were taken following the regulations of the fourth Hospital of Hebei Medical University. All male mice were housed and handled according to the guidelines of the local Animal Care and Use Committee at Hebei Medical University (approval number HEBMU-2011-09).

### Human tissue harvest

This study included (1) all patients with newly detected clinically intact (non-ruptured) aneurysms ($n = 22$), which underwent elective open AAA repair, and (2) those who were admitted for ruptured AAA ($n = 4$). AAA tissue specimens were obtained in the operating room from 26 male patients. Demographic and clinical characteristics of patients with AAA were presented in Table 1. Nonaneurysmal infrarenal aortic wall tissue specimens were also obtained from organ donors to serve as nonaneurysmal controls ($n = 8$). Each of the surgical patients gave informed signed consent before donating tissue. Aneurysm size was measured on preoperative CT angiograms.

### Animal study

ApoE$^{-/-}$ mice and Tgln-cre mice were purchased from Jackson Laboratory (Bar Harbor, ME), and Klf5-flox mice were a gift from Huajing Wan (Wuhan University, China). We generated mice with smooth muscle cell–specific deletion of Klf5 by crossing Klf5-flox mice and Tgln-cre mice (smcKlf5$^{-/-}$ mice) [58]. ApoE$^{-/-}$ smcKlf5$^{-/-}$ double-knockout mice were generated by crossing ApoE$^{-/-}$ and smcKlf5$^{-/-}$ mice. Homologous littermates with Klf5$^{+/+}$ phenotype as WT controls were used for experiments. Genotyping was performed by PCR.

To induce AAAs, we performed a mouse model of Ang II–induced AAA as previously described [13]. Digital photographs of the abdomens were taken to measure the maximum

external diameters. The abdominal arteries were harvested for analysis of RNA, morphology, and histology.

Mitochondrial division inhibitor, Mdivi-1, was purchased from Enzo life sciences (Plymouth meeting, PA). Mdivi-1 was given as intraperitoneal injection in the dose of 30 mg per kg body weight every other day for 2 weeks. Because Mdivi was dissolved in DMSO, we gave similar dilution of DMSO in the normal saline as vehicle control. At the end of the treatment, animals were humanely killed, then aortas were harvested for analysis as above.

Klf5 inhibitor, ML264, was purchased from MedChemExpress (CAS No. 1550008-55-3, US). All the mice were treated with ML246 (15 mg/kg) or the vehicle solution as the negative control, as previously described [59]. All the mice were injected intraperitoneally every two days for 14 or 28 days. At the end of the experiment, all animals were euthanized and aortas were harvested for analysis, as above.

## Blood pressure measurements

Systolic blood pressure was measured on a weekly basis using noninvasive tail cuff plethysmography. Pressure was obtained using a MC 4000 blood pressure analysis system for mice (Hatteras Instruments; Cary, NC) as reported previously [60]. Briefly, conscious mice were placed in a restrainer in a prewarmed chamber for 15–20 minutes before blood pressure examination. A pneumatic pulse transducer was placed on the tails of the mice, and the signals from it were collected and analyzed automatically using a data acquisition and analysis system. Each mouse underwent five successive rounds of measurements, and the mean of these values was recorded.

## Echocardiography

The cardiac function of all the mice was assessed by echocardiography using Vevo2100 system (VisualSonics, CA) with a 15-MHz linear transducer, as reported previously [60]. Mice were trained for 2–3 days before assessment of cardiac function to eliminate the variability of the measurement. All the echocardiography assessment was performed on conscious mice to eliminate the depressant effect of anesthesia on respiration and cardiac function. Diastolic measurements were made at the maximum left ventricle cavity dimension, whereas systolic parameters were measured during maximum anterior motion of the posterior wall.

## Generation of primary cells and cell culture

Mouse primary VSMCs were isolated from aortas of 25-g male mice anesthetized intraperitoneally with urethane as described previously [61]. Mouse primary VSMCs were cultured in low-glucose Dulbecco's Modified Eagle Medium (DMEM; Gibco Life Technologies, Rockville, MD) containing 100 units/mL of penicillin, 100 μg/mL of streptomycin, and 10% fetal bovine serum (GEMINI, USA) in a humidified incubator at 37˚C with 5% $CO_2$. The growth medium was replaced every 2 days, and the cells were passaged every 4 days at a ratio of 1 to 4 upon 80% confluence. Human aortic VSMCs (ScienCell, no. 6110) were routinely cultured in low-glucose DMEM. The culture method is similar to that of mouse primary VSMCs. Mdivi-1 (10 μmol/L; Enzo) was used to block mitochondrial fission.

## SA-β-gal staining

Cellular SA-β-gal activity was assayed using the Senescent Cells Staining Kit (Sigma, CS0030, USA) as previously described [62]. The SA-β-gal signals were analyzed using ImageJ software (NIH).

## Adenovirus expression vector and plasmid constructs

The expression plasmids of eIF5a and Mfn1 were created by the placement of human eIF5a and Mfn1 cDNA into the pcDNA3.1 vector. The 5′ promoter regions of human eIF5a, Atp5b, and Cox6a2 (−1,500 to +1 bp) were amplified by PCR and cloned into the pGL3-basic vector (Promega) in order to generate the eIF5a, Atp5b, and Cox6a2 promoter-reporter pGL3-eIF5a-luc, pGL3-Atp5b-luc, and pGL3-Cox6a2-luc. Truncated eIF5a luciferase reporters were generated by cloning the −1,379, −674, −287, and −89 to +1 regions of the eIF5a promoter into pGL3. Adenoviruses encoding Klf5 (Ad-Klf5) and control (Ad-Ctl), Ad-shKlf5, and Ad-U6 were entrusted to Invitrogen. The VSMCs were washed and incubated in serum-free medium and then were infected with the above adenovirus ($5 \times 10^9$ pfu/mL) for 36 hours, followed by Ang II treatment.

## siRNA transfection

siRNAs targeting human eIF5a (si-eIF5a), Mfn1 (si-Mfn1), and Drp1 (si-Drp1) and negative control were designed and synthesized by GenePharma (Shanghai, China). The siRNAs were transiently transfected into VSMCs using the Lipofectamine 2000 reagent (Invitrogen). Twenty hours following transfection, human VSMCs were treated with Ang II (100 ng/mL). Cells were then harvested and lysed for western blotting.

## qRT-PCR

Total RNA was extracted from VSMCs using TRizol Reagent, and the cDNA was synthesized using the reverse transcriptase kit (Invitrogen). Real-time PCR was performed using SYBR Green RT-PCR Kit (Invitrogen). The relative mRNA expression was normalized to GAPDH and calculated using the $2^{-\Delta\Delta Ct}$ formula, as previously described [63]. Sequence-specific primers used were presented in **Table 2**.

**Table 2. The primers for real-time PCR.**

| Primer | | Sequence (5′ to 3′) | Primer | | Sequence (5′ to 3′) |
|---|---|---|---|---|---|
| Klf5 | forward | ACCAGACGGCAGTAATGGACAC | eIF5a | forward | TGCCAAGGTCCATCTGGTT |
| | reverse | ATTGTAGCGGCATAGGACGGAG | | reverse | CAGCTGCCTCCTCTGTCATG |
| Drp1 | forward | TCACCCGGAGACCTCTCATT | eIF5a | forward | GATCCTGATCACGGTGCTGT |
| | reverse | TCTGCTTCCACCCCATTTTCT | (human) | reverse | GTCCAGCTTAGGACCGGG |
| Fis1 | forward | TCAGCCCCATCATGTGCTTT | Cdkn1a | forward | CAGAATAAAAGGTGCCACAGGC |
| | reverse | AGGAGAGGACCAGGAGTGAC | | reverse | CGTCTCCGTGACGAAGTCAA |
| Atp5b | forward | CCAGCAGATTTTAGCAGGTGAA | Mapk14 | forward | CCAAGCCATGAGGCAAGAAAC |
| | reverse | CTTTGGCTGGAGTCCCTCAC | | reverse | GGGTCGTGGTACTGAGCAAA |
| Pink1 | forward | AGTCCATTGGTAAGGGCTGC | Nfe2l2 | forward | GCAGGCTATCTCCTAGTTCTCC |
| | reverse | GAACCTGCCGAGATGTTCCA | | reverse | ATCAAATCCATGTCCTGCTGGGA |
| Mtfr1 | forward | GCATGCAATCGATTAACAGCCA | GAPDH | forward | AAGGTGAAGGTCGGAGTC |
| | reverse | CAAAAGACGCCACTGCATCC | | reverse | GATTTTGGAGGGATCTCG |
| Mfn1 | forward | GGGTGATAGTTGGAGCGGAG | Cox4i1 | forward | CTTGGACGGCGGAATGTTGG |
| | reverse | ATCGCCTTCTTAGCCAGCAC | | reverse | TCAGCGTAAGTGGGGAAAGC |
| β-actin | forward | AAATCGTGCGTGACATCAAAGA | Cox6a2 | forward | GCATCCGAACCAAGCCCT |
| | reverse | GGCCATCTCCTGCTCGAA | | reverse | GCGTGTCTGCTGAGACATCA |
| NOX1 | forward | TTCCCTGGAACAAGAGATGG | NOX4 | forward | CTGGAAGAACCCAAGTTCCA |
| | reverse | GACGTCAGTGGCTCTGTCAA | | reverse | CGGATGCATCGGTAAAGTCT |

In addition to the specific primers for human marked in brackets, the rest of primer sequences are specific for mouse.

Abbreviations: Atp5b, ATP synthase subunit β; Cdkn1a, cyclin dependent kinase inhibitor 1a; Cox4i1, cytochrome c oxidase subunit 4i1; Cox6a2, cytochrome c oxidase subunit 6a2; Drp1, dynamin-related protein 1; eIF5a, eukaryotic translation initiation factor 5a; Fis1, fission mitochondrial 1; GAPDH, glyceraldehyde-3-phosphate dehydrogenase; Klf5, Krüppel-like factor 5; Mapk14, mitogen-activated protein kinase 14; Mfn1, mitofusin 1; Mtfr1, mitochondrial fission regulator 1; Nfe2l2, nuclear factor, erythroid 2 like 2; NOX, NAPDH oxidase; Pink1, PTEN-induced kinase 1

## qRT-PCR for mitochondrial DNA content

Mitochondrial DNA (mtDNA) copy number was quantified by qPCR using the mtDNA Copy Number Kit (MCN3) (Detroit R&D, MI). DNA was extracted from aortic tissues and cultured VSMCs using the DNA isolation kit (Qiagen). The total DNA concentration was determined using a NanoDrop 1000 (Thermo Fisher Scientific). MtDNA levels were quantitated by normalizing the mitochondrial gene (cytochrome b) to the nuclear gene (GAPDH). Evaluation of mtDNA content in tissue homogenate and cultured cells was performed by qPCR, as previously described [64].

## Microarray analysis

Total RNA was extracted by using TRIzol extraction method and a NucleoSpin RNA II kit (Macherey Nagel, Duren, Germany), from WT ($n = 3$) and smcKlf5$^{-/-}$ ($n = 3$) mouse aortas at day 28. mRNA profile was assayed using the array hybridization as previously described. All chip data used in this paper are from NCBI, and the accession number from NCBI is GSE148841. All gene-level files were imported into Agilent GeneSpring GX software (version 12.1) for further analysis. Genes in 6 samples have values greater than or equal to lower cutoff: 100.0 ("All Targets Value") were chosen for data analysis. GO Analysis was applied to determine the roles of these differentially expressed genes played in these biological GO terms.

## RNA-Seq

RNA extraction of Ad-GFP–and Ad-Klf5–infected mouse VSMCs was performed and sequenced on an Illumina HiSeq 2000. Illumina sequencing libraries were prepared according to the TruSeq RNA Sample Preparation Guide following the manufacturer's instructions. Libraries were sequenced using $1 \times 58$ bp single-end reads, with two indexed samples per lane, yielding about 32.5 million reads per sample. After alignment of the sequencing reads to the human genome, counts for each gene were computed for each sample by use of the HTSeq software (version v0.5.3p3). All RNA-Seq data used in this paper are from NCBI, and the accession number is GSE148765.

## Immunoblot analysis

Lysates from VSMCs were prepared and separated by SDS-PAGE, transferred to Immobilon-P membranes (Millipore), and incubated with specific antibodies. Western Lightning plus-ECL (PerkinElmer) was used for detection. Klf5 antibody (GTX103289, GeneTex) was from BD Biosciences. Antibodies of eIF5a (17069-1-AP), Fis1 (10956-1-AP), Mfn1 (13798-1-AP), Drp1 (12957-1-AP), Mftr1 (bs-7632R), MMP2 (10373-2-AP), MCP-1 (66272-1-Ig), NOX1 (17772-1-AP), Mff (17090-1-AP), Mid51 (20164-1-AP), Mid49 (16413-1-AP), Mfn2 (12186-1-AP), OPA (27733-1-AP), PGC-1α (ab54481, Abcam), and mtTFA (ab131607, Abcam) were from Proteintech (USA). β-actin (ab8226) was from Abcam (Cambridge, UK). Band intensities were quantified with the ImageJ software (NIH).

## Luciferase assay

Human embryonic kidney 293A cells were maintained as previously described [65]. A total of $3 \times 10^4$ VSMCs were seeded into each well of a 24-well plate and grown for 24 hours prior to transfection with reporter plasmids and the control pTK-RL plasmid. VSMCs were transfected using Lipofectamine 2000 reagent (Invitrogen) according to the manufacturer's instructions. Luciferase assays were performed after 24 hours using a dual luciferase assay kit (Promega). Specific promoter activity was expressed as the relative ratio of firefly luciferase activity to

Renilla luciferase activity. All promoter constructs were evaluated in a minimum of three separate wells per experiment.

## ChIP assay

VSMCs were cross-linked with 1% formaldehyde for 15 minutes, lysed as previously described, and then sonicated to an average size of 400–600 bp [66]. The DNA fragments were immunoprecipitated overnight with anti-Klf5 or anti-mouse immunoglobulin G (IgG) (as a negative control). After the reversal of cross-linking, the eIF5a-flanking genomic region containing Klf5-binding sites was amplified by PCR; the forward and reverse primers used were as follows: Primer 1: forward: 5′-ccagacagggcaaaggctcc-3′, reverse: 5′-cggtgctacggctgcccctgc-3′; Primer 2: forward: 5′-gcgaggtgagagcgggcagg-3′, reverse: 5′-ggccccaaccgtcacccctc-3′; and Primer 3: forward: 5′-cgttgagacccaggcgtttc-3′, reverse: 5′-ttccaaccagtcggcaatgc-3′.

## Oligonucleotide pull-down assay

Oligonucleotide pull-down assay was undertaken as described previously [66]. Oligonucleotides containing the WT or mutant TCE site (site-1–3) sequence in the human eIF5a promoter with biotin added to their 5′ end were as follows: site-1 (WT): biotin-5′-gcgcggccacgtgaggtggg ggaggggggct-3′ (forward), biotin-5′-agcccccctcccccacctcacgtggccgcgc-3′ (reverse); mutant site-1: biotin-5′-gcgcggccacgtgagacaggggagggggct-3′ (forward), biotin-5′-agcccccctcccctgtctcacgtggcc gcgc-3′ (reverse); site-2 (WT): biotin-5′-gccggccgggtgacgttggagcga-3′ (forward), biotin-5′-tcg ctccaacgtcacccggccggc-3′ (reverse); mutant site-2: biotin-5′-gccggccggacaacgttggagcga-3′ (forward), biotin-5′-tcgctccaacgttgtccggccggc-3′ (reverse); and site-3 (WT): biotin-5′-gctgtcgctcaa ggggagggaagaagaggt-3′ (forward), biotin-5′-acctcttcttccctcccccttgagcgacagc-3′ (reverse).

## Co-immunoprecipitation assay

Co-immunoprecipitation was performed as described previously [67]. The cell lysates were immunoprecipitated with anti-eIF5a antibodies or anti-Mfn1, anti-Drp1, anti-Fis1, and anti-Mftr1 antibodies for 1 hour at 4°C, followed by incubation with protein A agarose overnight at 4°C. The precipitates were collected by centrifugation at 12,000*g* for 1 minute at 4°C and washed 5 times with cold RIPA buffer before immunoblots using antibodies against eIF5a, Mfn1, Drp1, Fis1, and Mftr1.

## Proximity ligation assay

The experiment was performed as described previously [13]. VSMCs were grown on cell culture inserts and incubated with 4% paraformaldehyde for 10 minutes. Proximity ligation assay was performed using the Rabbit PLUS and Mouse MINUS Duolink in situ proximity ligation assay kits with anti-eIF5a (mouse) and anti-Mfn1 (rabbit; OLINK Bioscience, Uppsala, Sweden) according to the manufacturer's protocol. Subsequently, slides were dehydrated, air-dried, and embedded in DAPI-containing mounting medium.

## Immunohistochemistry

CD45 expression by IHC was examined in formalin-fixed and paraffin-embedded aortic tissues from mouse AAA model using a rabbit polyclonal anti-mouse CD45 antibody (ab10558, abcam) at 1:200 dilution and 4°C overnight, after further incubation with anti-rabbit peroxidase-labeled polymer. Immunohistochemical staining was visualized by use of a diaminobenzidine kit (Zhongshan Goldenbridge Biotechnology, China) according to the manufacturer's

instructions. Sections were counterstained with hematoxylin to visualize nucleus. Images were acquired using a Leica microscope (Leica DM6000B, LAS V.4.3, Switzerland).

## Cell immunofluorescence

Immunofluorescence staining was performed as described previously [13]. Cells were permeabilized with 0.1% Triton X-100 in PBS and then blocked with 0.1% Triton X-100 and 5% BSA in PBS for 1 hour, washed, and incubated overnight with the primary antibodies anti-eIF5a (1:50, sc-390062, Santa), anti-Drp1 (1:100, 12957-1-AP, Proteintech), anti-Mfn1 (1:100, 13798-1-AP, Proteintech), anti-Fis (1:100, 10956-1-AP, Proteintech), and anti-Mftr1 (1:50, bs-7632R, Bioss) at 4˚C. Antibodies conjugated to Alexa Fluor 405, 488, and 568 were used as secondary antibodies. Mitochondrial was visualized with Mitotracker green and nuclei were stained with DAPI. Images were captured by confocal microscopy (DM6000 CFS, Leica) and processed by LAS AF software. All images of primary cells were taken with 63×NA 0.75 objectives. One hundred cells were counted for each assay.

## Measurement of total and mtROS levels

The ROS measurement was performed as previously described [68]. ROS level was evaluated by analyzing the fluorescence intensity that resulted from DHE (Invitrogen) and MitoSOX (Invitrogen) staining. In brief, frozen mouse aortas were cut into 4-μm sections. Serial aorta sections were stained with 5 μM DHE or MitoSOX at 37˚C for 30 minutes and then measured by fluorescence microscopy. The treated VSMCs were loaded with 5 μM DHE or mitoSOX at 37˚C for 30 minutes in the dark and then measured by flow cytometer (BD Biosciences, San Jose, CA). The MitoSOX fluorescence intensities were analyzed using FlowJo software.

## ATP assay

ATP levels were determined using the Adenosine 5′-triphosphate (ATP) Bioluminescent Assay Kit (Sigma) following the manufacturer's instructions. Luminescence was recorded using a luminometer (Molecular Probes).

## Measurement of Δψm

Δψm was assessed in VSMCs using the JC-1 probe (Beyotime) according to the manufacturer's protocols. Following incubation and treatment, cells were incubated with JC-1 staining solution for 20 minutes at 37˚C. Then cells were collected and analyzed using a flow cytometer (Elite Epics; Beckman Coulter) with excitation at 488 nm, and emission at 529 nm (monomer form of JC-1, green) and at 590 nm (aggregate form of JC-1, red). The Δψm of VSMCs in each group was calculated as the fluorescent ratio of red to green.

## Measurement of mitochondrial respiratory chain complex activities

Mitochondria were isolated from VSMCs using Mitochondria Isolation Kits (Solarbio, China) according to the manufacturer's protocol. The activities of complexes I, II, III, and IV were measured with the MitoProfile Rapid Microplate Assay Kit (Abcam) according to the manufacturer's instructions. Complexes I, II, III, and IV were immune-captured in the wells in the microplates, and enzymatic activity was measured with a kinetic colorimetric assay kit (Thermo Fisher Scientific). The total and mitochondrial protein concentrations were determined using the BCA assay (Beyotime).

## TUNEL assay

A terminal deoxynucleotidyl-transferase-mediated dUTP nick end labeling (TUNEL) assay kit (FITC, ab66108) was used to identify apoptosis cells in aorta tissues following the manufacturer's instructions. The FITC-labeled TUNEL-positive cells in aorta tissues were imaged under a fluorescent microscopy by using 488-nm excitation and 530-nm emission. Apoptosis of human VSMCs was detected by flow cytometry. In brief, the cells were harvested by trypsinization and washed twice with cold PBS. Next, the cells were centrifuged and resuspended in the binding buffer at a density of $1.0 \times 10^6$ cells/mL. The cells were incubated with 5 μL of annexin V-FITC and 5 μL of PI for 15 minutes at room temperature in the dark. All samples were subject to flow cytometry (FC500 MPL Beckman) and analyzed with the FlowJo software.

## Transmission electron microscopy

Samples were processed following standard protocol, including dehydration, embedding, and sectioning and then examined and captured under a Hitachi JEM-1400 transmission electron microscope (Hitachi, Tokyo, Japan). The number of mitochondria was quantified by counting the total number of mitochondria per 100 μm$^2$ of cell surface. Mitochondrial mean diameter was calculated as the average diameter per mitochondrion; 200 mitochondria per group were randomly selected. Mitochondrial major and minor axes lengths were quantified using NIH ImageJ software.

## Semi-quantification by ImageJ

For each experiment, three or more fields of view were taken as fluorescent images for each group. Fluorescent intensity in each image was semiquantitated with ImageJ and averaged. Results were displayed as mean fluorescence intensity.

## Statistical analysis

All data are displayed as representative or results from multiple independent experiments. Data comparisons were performed with one-way analysis of variance test. Two-sided $P$ values of $<0.05$ were considered significant and denoted with 1, 2, or 3 asterisks when lower than 0.05, 0.01, or 0.001, respectively. No randomization or blinding was used.

## Supporting information

**S1 Fig. Analysis of human aneurysmal and normal aortic tissues.** (A) Representative 3D volume-rendered image of abdomen from the *normal* human abdominal aorta and unruptured and ruptured aneurysms. (B, C) Representative HE-stained and Masson-stained sections from the normal human abdominal aorta and unruptured and ruptured aneurysms. Scale bars = 500 and 50 μm. HE, hematoxylin–eosin.
(TIF)

**S2 Fig. The effect of ML264 (Klf5 inhibitor) on aneurysm formation induced by Ang II infusion in the early and late stages.** Representative ultrasound imaging of mouse AAA models induced by Ang II infusion for 14 and 42 days in ApoE$^{-/-}$ mice injected intraperitoneally with ML264 every two days for 14 and 42 days. Ang II, angiotensin II; Klf5, Krüppel-like factor 5.
(TIF)

**S3 Fig. ERK signaling and expression of β-arrestin2 in Ang II–induced mouse AAA models as well as in WT or Klf5$^{-/-}$ VSMCs.** (A) Systolic blood pressure measured by tail cuff method

in conscious mice following 0, 14, 21, 28, 42 days of Ang II infusion. (B) The expression of β-arrestin2 and ERK1/2 was analyzed by western blotting in Ang II–injured mouse aortas for 28 and 42 days. β-actin was used as a loading control. **$P < 0.01$ and ***$P < 0.01$ versus 0 day. (C) The expression of β-arrestin2 and ERK1/2 was analyzed by western blotting in WT and Klf5$^{-/-}$ VSMCs. *$P < 0.05$ and **$P < 0.01$ versus WT. For numerical raw data, please see S1 Data. For raw immunoblots, please see S1 Blots. AAA, abdominal aortic aneurysm; Ang II, angiotensin II; ERK, extracellular signal–regulated kinase; Klf5, Krüppel-like factor 5; VSMC, vascular smooth muscle cell; WT, wild-type.
(TIF)

**S4 Fig. Young (3 months) or old (18 months) WT and smcKlf5$^{-/-}$ mice were infused with Ang II for 28 days.** (A) Representative photographs and quantitative analysis of SA-β-gal–stained aortas from WT and smcKlf5$^{-/-}$ mice. Scale bars = 5 mm; $n = 5$ per group, *$P < 0.05$ and **$P < 0.01$ versus WT or young smcKlf5$^{-/-}$ mouse. (B) Representative images of SA-β-gal–stained transverse sections of abdominal aortas from WT and smcKlf5$^{-/-}$ mice. Blue staining indicates SA-β-gal–positive stained cells, and cytoplasm and extracellular matrix were counterstained using HE. Scale bars = 50 μm. For numerical raw data, please see S1 Data. Ang II, angiotensin II; HE, hematoxylin–eosin; SA-β-gal, senescence-associated β-galactosidase; WT, wild-type.
(TIF)

**S5 Fig. Cardiac function assessed by echocardiography in Ang II–infused young (3 months) or old (18 months) WT and smcKlf5$^{-/-}$ mice.** (A) Ejection fraction, (B) shortening fraction, (C) left ventricular dimension at systole, (D) left ventricular dimension at diastole. *$P < 0.05$, **$P < 0.01$ versus WT. $n = 6$ for each group. For numerical raw data, please see S1 Data. WT, wild-type.
(TIF)

**S6 Fig. Representative TUNEL- and DAPI-stained sections from the abdominal aortas of young and old WT and smcKlf5$^{-/-}$ mice following 28 days of Ang II infusion.** Graphical data represent the percentage of apoptotic cells (green)/the total number of nucleated cells (blue). $n = 3$ in each group, *$P < 0.05$ and **$P < 0.01$ versus WT or young mice. Scale bars = 50 μm. For numerical raw data, please see S1 Data. Ang II, angiotensin II; WT, wild-type.
(TIF)

**S7 Fig. VSMCs were stimulated with Ang II (100 nmol/L) for the indicated times.** Representative immunofluorescent images of Ki67 (green) and phalloidin (red) staining of VSMCs treated with Ang II. Scale bars = 5 μm. Ang II, angiotensin II; VSMC, vascular smooth muscle cell.
(TIF)

**S8 Fig. The expression of eIF5a, Fis1, Pink1, Drp1, Mfn1, and Mtfr1 and the analysis of mitochondrial morphology.** (A) Representative western blot image of eIF5a, Fis1, Pink1, Drp1, Mfn1, and Mtfr1 in Klf5$^{-/-}$ VSMCs infected or not with Ad-Klf5. (B) Representative western blot image of eIF5a, Fis1, Pink1, Drp1, Mfn1, and Mtfr1 in human VSMCs infected with Ad-Klf5 and Ad-Ctl or Ad-shKlf5. (C) MitoTracker Red–stained mitochondria in VSMCs infected with indicated constructs. Right: the percentage of cells containing fused and fragmented mitochondria was quantified from more than 100 cells. Scale bars = 10 μm. Data represent mean ± SEM, **$P < 0.01$ versus Ad-Ctl; #$P < 0.05$ and ##$P < 0.01$ versus Ad-shKlf5. For numerical raw data, please see S1 Data. Ad-Ctl, adenoviruses encoding control; Ad-Klf5,

adenoviruses encoding Klf5; Ad-shKlf5, adenoviruses encoding small hairpin Klf5; Drp1, dynamin-related protein 1; eIF5a, eukaryotic translation initiation factor 5a; Fis1, fission mitochondrial 1; Klf5, Krüppel-like factor 5; Mfn1, mitofusin 1; Mtfr1, mitochondrial fission regulator 1; Pink1, PTEN-induced kinase 1; VSMC, vascular smooth muscle cell.
(TIF)

**S9 Fig. The correlation of the mitochondrial dynamics–related genes with Klf5 in mouse VSMCs.** (A) Representative western blot image of Nfe2l2, Mapk14, Cdkn1a, Tmx2, Atp5b, and Cox6a2 in WT and Klf5$^{-/-}$ VSMCs. Right: Band intensities that were measured and normalized to β-actin ($n$ = 3). Data represent the mean ± SD. $^*P < 0.05$ and $^{**}P < 0.01$ versus WT. (B) Representative western blot image of Nfe2l2, Mapk14, Cdkn1a, Tmx2, Atp5b, and Cox6a2 in Ad-Klf5– and Ad-Ctl–infected mouse VSMCs. Right: Band intensities that were measured and normalized to β-actin ($n$ = 3). $^*P < 0.05$ versus Ad-Ctl. (C,D) Cells (293A) were transfected with the reporter directed by the Atp5b (C) or Cox6a2 (D) promoter, and luciferase activity was measured. Data represent the relative eIF5a or Cox6a2 promoter activity normalized to pRL-TK activity. For numerical raw data, please see S1 Data. Ad-Ctl, adenoviruses encoding control; Ad-Klf5, adenoviruses encoding Klf5; Atp5b, ATP synthase subunit β; Cdkn1a, cyclin dependent kinase inhibitor 1a; Cox6a2, cytochrome c oxidase subunit 6A isoform 2; eIF5a, eukaryotic translation initiation factor 5a; Klf5, Krüppel-like factor 5; Mapk14, mitogen-activated protein kinase 14; Nfe2l2, nuclear factor, erythroid 2 like 2; pRL-TK, thymidine kinase promoter-Renilla luciferase reporter plasmid; Tmx2, thioredoxin-related transmembrane protein 2; VSMC, vascular smooth muscle cell; WT, wild-type.
(TIF)

**S10 Fig. Klf5 deficiency leads to mitochondrial dyfunction in VSMCs.** (A-E) Analysis of ATP content (A), relative ΔΨm (B), and activities of complexes I, II, and IV (B-D) in WT and Klf5$^{-/-}$ VSMCs as well as in Ad-Klf5–infected Klf5$^{-/-}$ VSMCs. $^*P < 0.05$ and $^{**}P < 0.01$ versus WT. (F-I) Mouse VSMCs were stimulated with Ang II (100 nmol/L) for the indicated times, and then the activities of complexes I, II, III, and IV (F), ATP content (G), the expressions of PGC-1α and mtTFA (H), and the copy number of mtDNA (I) were analyzed. $^*P < 0.05$, $^{**}P < 0.01$, and $^{***}P < 0.001$ versus 0 day. For numerical raw data, please see S1 Data. For raw immunoblots, please see S1 Blots. Ad-Klf5, adenoviruses encoding Klf5; Ang II, angiotensin II; Klf5, Krüppel-like factor 5; mtDNA, mitochondrial DNA; mtTFA, mitochondrial transcription factor A; PGC-1α, peroxisome proliferative activated receptor, gamma, coactivator 1 alpha; VSMC, vascular smooth muscle cell; WT, wild-type; ΔΨm, mitochondrial membrane potential.
(TIF)

**S11 Fig. Interaction of eIF5a with mitochondrial dynamics–related proteins.** (A-C) Reciprocal co-immunoprecipitation assay for interaction between eIF5a and Drp1 (A), Fis1 (B), or Mtfr1 (C) in VSMCs treated with Ang II (100 nmol/L) for the indicated times. IgG was used as a negative control. For raw immunoblots, please see S1 Blots. Ang II, angiotensin II; Drp1, dynamin-related protein 1; eIF5a, eukaryotic translation initiation factor 5a; Fis1, fission mitochondrial 1; IgG, immunoglobulin G; Mtfr1, mitochondrial fission regulator 1; VSMC, vascular smooth muscle cell.
(TIF)

**S12 Fig. eIF5a affects the expression of mitochondrial dynamics–related genes.** (A) The percentage of cells containing fused and fragmented mitochondria in Fig 6C. $^*P < 0.05$ versus pcDNA3.1. (B) Relative fluorescence intensity of Mfn1 in Fig 6C. Data represent the mean ± SD. $^{**}P < 0.01$ versus pcDNA3.1. (C) Representative western blot image of eIF5a, Fis1, Mff, Mid51, Mid49, Mtfr1, Mfn1, Mfn2, and OPA1 in mouse VSMCs transfected with

pcDNA3.1 or pcDNA3.1-eIF5a. Right: Band intensities that were measured and normalized to β-actin. Data represent the mean ± SD. $*P < 0.05$ and $**P < 0.01$ versus pcDNA3.1. For numerical raw data, please see S1 Data. For raw immunoblots, please see S1 Blots. eIF5a, eukaryotic translation initiation factor 5a; Fis1, fission mitochondrial 1; Mff, mitochondrial fission factor; Mfn1, mitofusin 1; Mfn2, mitofusin 2; Mid49, mitochondrial dynamics protein 49; Mid51, mitochondrial dynamics protein 51; Mtfr1, mitochondrial fission regulator 1; OPA1, Optic atrophy type 1; VSMC, vascular smooth muscle cell.
(TIF)

**S13 Fig. Analysis of relative fluorescence intensity.** (A, B) Statistic analysis of relative fluorescence intensity of Mfn1 and eIF5a in Fig 6F (A) and Fig 6G (B), respectively. Data represent the mean ± SD. $**P < 0.01$ versus pcDNA3.1. For numerical raw data, please see S1 Data. eIF5a, eukaryotic translation initiation factor 5a; Mfn1, mitofusin 1.
(TIF)

**S14 Fig. Mitochondrial morphology assay.** MitoTracker green staining detects the mitochondrial morphology in mouse VSMCs transfected with expression plasmids encoding Klf5, eIF5a, or Mfn1. Down: The percentage of cells containing fused and fragmented mitochondria was quantified from more than 100 cells. Scale bars = 10 μm. Data represent mean ± SEM, $*P < 0.05$, $**P < 0.01$, and $***P < 0.001$ versus pcDNA3.1. For numerical raw data, please see S1 Data. eIF5a, eukaryotic translation initiation factor; Klf5, Krüppel-like factor 5; Mfn1, mitofusin 1; VSMC, vascular smooth muscle cell.
(TIF)

**S15 Fig. ATP assay.** ATP content in mouse VSMCs infected or transfected with Ad-Klf5 or pcDNA3.1-eIF5a and then treated with Ang II for 5 days. Mouse VSMCs were infected or transfected with Ad-Klf5 or pcDNA3.1-eIF5a and then treated with Ang II for 5 days. ATP levels were determined using the Adenosine 5′-triphosphate (ATP) Bioluminescent Assay Kit (Sigma) following the manufacturer's instructions. Data represent mean ± SEM, $*P < 0.05$ versus Ad-Ctl or pcDNA3.1. For numerical raw data, please see S1 Data. Ad-Ctl, adenoviruses encoding control; Ad-Klf5, adenoviruses encoding Klf5; Ang II, angiotensin II; eIF5a, eukaryotic translation initiation factor 5a; VSMC, vascular smooth muscle cell.
(TIF)

**S16 Fig. Effect of eIF5a and Mfn1 expression on mitochondrial morphology.** (A,B) Mouse VSMCs were transfected with si-NC, si-eIF5a, pcDNA3.1, or pcDNA3.1-eIF5a (A), as well as with si-NC, si-Mfn1, pcDNA3.1, or pcDNA3.1-Mfn1 (B) for 36 hours. Crude proteins were extracted and then subjected to western blotting with anti-eIF5a or anti-Mfn1 antibodies. β-actin was used as a loading control. (C) Mouse VSMCs were co-transfected with pcDNA3.1 or pcDNA3.1-eIF5a and si-NC or si-Mfn1 for 36 hours. Co-localization of eIF5a with Mfn1 and Mitotracker was detected by confocal microscopy. Scale bars = 10 μm. The percentage of cells containing fragmented and fused mitochondria was quantified from more than 300 cells. Data are expressed as mean ± SEM. $**P < 0.01$ or $***P < 0.001$ versus si-NC. For numerical raw data, please see S1 Data. For raw immunoblots, please see S1 Blots. eIF5a, eukaryotic translation initiation factor 5a; Mfn1, mitofusin 1; si-eIF5a, short interfering RNA targeting eIF5a; si-Mfn1, short interfering RNA targeting Mfn1; si-NC, short interfering RNA negative control; VSMC, vascular smooth muscle cell.
(TIF)

**S17 Fig. Klf5 and eIF5a suppress Ang II–induced expression of NOX1 in VSMCs.** (A-D) Mouse VSMCs were infected or transfected with Ad-Klf5 and Ad-shKlf5 (A,B) or

pcDNA3.1-eIF5a and si-eIF5a (C,D) and then treated with Ang II for the indicated times. NOX1 and NOX4 mRNA expression was quantified using qRT-PCR, and normalized to GAPDH and expressed as fold increase over day 0. (E,F) Klf5-overexpressing or knocking down (E) and eIF5a-overexpressing or knocking down (F) mouse VSMCs were treated with Ang II for 24 hours. Crude proteins were extracted and then subjected to western blotting with anti-NOX1 antibody. β-actin was used as a loading control. Right: Band intensities that were measured and normalized to β-actin. Data represent the mean ± SD. $^*P < 0.05$ and $^{**}P < 0.01$ versus 0 day, $^#P < 0.05$ versus Ad-Ctl and pcDNA3.1 at the same day or at the Ang II treatment. For numerical raw data, please see S1 Data. For raw immunoblots, please see S1 Blots. Ad-Ctl, adenoviruses encoding control; Ad-Klf5, adenoviruses encoding Klf5; Ad-shKlf5, adenoviruses encoding small hairpin Klf5; Ang II, angiotensin II; eIF5a, eukaryotic translation initiation factor 5a; GAPDH, glyceraldehyde-3-phosphate dehydrogenase; Klf5, Krüppel-like factor 5; NOX1, NAPDH oxidase 1; qRT-PCR, quantitative real-time PCR; si-eIF5a, short interfering RNA targeting eIF5a; VSMC, vascular smooth muscle cell.
(TIF)

**S18 Fig. NAC suppresses Ang II–induced mitochondrial fission in mouse VSMCs.** (A) Mouse VSMCs pretreated with NAC (500 μM) were treated or not with Ang II for the indicated times. Quantification of MitoSox fluorescence of VSMCs by FACS analysis using Mitosox Green dye. (B) NAC-pretreated VSMCs were stimulated with Ang II for 1 or 5 days, and then subjected to MitoTracker Green staining to visualize mitochondrial morphology. Right: The percentage of cells containing fragmented and fused mitochondria was quantified from more than 100 cells. All data represent the mean ± SEM. $^*P < 0.05$ and $^{**}P < 0.01$ versus 0 day, $^#P < 0.05$ and $^{##}P < 0.01$ versus untreated with NAC at the same day. For numerical raw data, please see S1 Data. Ang II, angiotensin II; FACS, flow analysis of cytosorting; NAC, N-Acetyl-L-cysteine; VSMC, vascular smooth muscle cell.
(TIF)

**S19 Fig. The effect of Mdivi-1 administration on cell apoptosis in vivo.** Mdivi-1 (1.2 mg/kg) was administrated to mice intraperitoneally 15 minutes before the onset of Ang II perfusion and then injected once a week for 4 weeks. Representative TUNEL-stained (green) and DAPI-stained (blue) slices show cell apoptosis in the abdominal aortas of ApoE$^{-/-}$ and ApoE$^{-/-}$ smcKlf5$^{-/-}$ mice. Scale bars = 50 μm. Ang II, angiotensin II; Mdivi-1, mitochondrial division inhibitor 1.
(TIF)

**S1 Data. Numerical raw data.** All numerical raw data are combined in a single Excel file, "S1_Data.xlsx". This file consists of several spreadsheets. Each spreadsheet contains the raw data of one subfigure.
(XLSX)

**S1 Blots. Raw images.** The file "S1_Blots.pdf" covers all uncropped western blot images, including size standards and descriptions.
(PDF)

## Acknowledgments

We are grateful to Dr. Xi-le Bi and Jun-jian Zhao for help in obtaining AAA samples.

## Author Contributions

**Conceptualization:** Dong Ma, Xin-hua Zhang, Qiang Li.

**Data curation:** Dong Ma, Yong-bo Zhao, Xiao Liu, Xin-hua Zhang, Qiang Li.

**Formal analysis:** Dong Ma, Xiao Liu, Xin-hua Zhang.

**Funding acquisition:** Dong Ma, He-liang Liu, Jin-kun Wen.

**Investigation:** Dong Ma, Yong-bo Zhao, Xiao Liu, Qiang Li, Wei-bo Shi.

**Methodology:** Dong Ma, Bin Zheng, Xin-hua Zhang, Qiang Li.

**Project administration:** Dong Ma, Bin Zheng, He-liang Liu, Jin-kun Wen.

**Resources:** Dong Ma, He-liang Liu, Yong-bo Zhao, Xiao Liu, Wei-bo Shi.

**Supervision:** Toru Suzuki, Jin-kun Wen.

**Validation:** Qiang Li, Toru Suzuki.

**Visualization:** Xin-hua Zhang, Qiang Li, Toru Suzuki.

**Writing – original draft:** Dong Ma.

**Writing – review & editing:** Jin-kun Wen.

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
