## [Editor Report · Decision Letter 0]

2 Jan 2020

Dear Dr Ma, 

Thank you for submitting your manuscript entitled "eIF5a r eduction elicited by Klf5 downregulation leads to cell senescence   through inducing mitochondrial fission in vascular smooth muscle cells" for consideration as a Research Article by PLOS Biology.

Your manuscript has now been evaluated by the PLOS Biology editorial staff as well as by an academic editor with relevant expertise and I am writing to let you know that we would like to send your submission out for external peer review.

Please re-submit your manuscript within two working days, i.e. by Jan 06 2020 11:59PM.

***Please be aware that, due to the voluntary nature of our reviewers and academic editors, manuscripts may be subject to delays due to their limited availability during the holiday season. Please also note that the journal office will be closed entirely 21st- 29th December inclusive, and 1st January 2020. Thank you for your patience.***

Kind regards,

Ines

--

Ines Alvarez-Garcia, PhD

Senior Editor

PLOS Biology

Carlyle House, Carlyle Road

Cambridge, CB4 3DN

+44 1223–442810

---

## [Decision Letter · Decision Letter 1]

5 Feb 2020

Dear Dr Ma,

Thank you very much for submitting your manuscript "eIF5a reduction elicited by Klf5 downregulation leads to cell senescence through inducing mitochondrial fission in vascular smooth muscle cells" for consideration as a Research Article at PLOS Biology. Your manuscript has been evaluated by the PLOS Biology editors, an Academic Editor with relevant expertise, and by three independent reviewers.

As you will see, the reviewers feel that the findings of the manuscript are novel and significant for the field, however they also think that several experiments need to be performed to confirm some of the findings. In addition, the reviewers ask for several controls and clarifications that are crucial to consider the paper further for publication.

In light of the reviews (attached below), we will not be able to accept the current version of the manuscript, but we would welcome re-submission of a revised version that takes into account the reviewers' comments. We cannot make any decision about publication until we have seen the revised manuscript and your response to the reviewers' comments. Your revised manuscript is also likely to be sent for further evaluation by the reviewers.

We expect to receive your revised manuscript within 3 months, however we are aware of the difficult situation in China at the moment, so please let us know if you need more time for the revision.

**IMPORTANT - SUBMITTING YOUR REVISION**

*Re-submission Checklist*

*Published Peer Review*

*PLOS Data Policy*

*Blot and Gel Data Policy*

Sincerely,

Ines

--

Ines Alvarez-Garcia, PhD

Senior Editor

PLOS Biology

Carlyle House, Carlyle Road

Cambridge, CB4 3DN

+44 1223–442810

Reviewers’ comments

Rev. 1:

This study examined the function of Krüppel-like factor 5 (Klf5) in abdominal aortic aneurysm (AAA). The authors found that Klf5 deficiency attenuates, while overexpression promotes mitochondria fusion. Mechanistically, the authors proposed that loss of Klf5 upregulates fission-related gene expression, which leads to enhanced ROS generation and vascular smooth muscle cell senescence. The findings are very interesting. However, more work needs to be done to reveal the role Klf5 plays in mitochondrial dynamics and function, as well as reveal its contribution to AAA, especially ruptured AAA.

1. Please divide the patients into "ruptured" and "unruptured" groups and report separately. Are these two groups comparable in terms of demographic and clinical characteristics?

2. Fig. 1h, there are Klf5 positive cells that are SMA negative, what are those cells?

3. What is the cause of death in the smcKlf5−/− mice? Rupture of aortic aneurysm or cardiac problems? How is their cardiac function after Ang II treatment?

4. The authors showed that one gene, elF5a, is the direct target of Klf5. How about the rest of the genes identified by RNAseq that were differentially expressed in Klf5 deficient or overexpressed cells? The expression of these genes were changed at the transcriptional level, if they were not regulated by Klf5, what is the direct regulator?

5. What is the cause of mitochondria fission when Klf5 is lost? Is it because that the fusion-related genes are repressed? Or is it because that mitochondrial function is impaired due to the downregulation of ATP production, and the changes in the expression of mitochondria dynamics-related genes are secondary ?

6. What is the physiological significance of the interaction between elF5a and Mfn1? The results suggested that overexpression of either one was sufficient to rescue the mitochondrial morphology, then is their interaction necessary?

7. In the wild type cells after 5 days of Ang II treatment, the mitochondria in both wild type and Klf5-/- cells were fragmented, while after 1 day of treatment, the morphology of the mitochondria in wild type and KO were different. Moreover, Klf5 was recruited to elF5a promoter after 24 hr of Ang II (Fig. 5g and h). Therefore, the rescue experiment, which meant to demonstrate the effects of Klf5 in AngII stimulated mitochondrial fission, should also be done after 1 day of treatment, and see if enforced expression of Klf5, elF5a or Mfn1 could reverse the fission phenotype in the KO cells.

8. How does the enhanced mitochondrial fission in VSMCs contribute to the exacerbated AAA outcome/rupture?

Minor:

1. In file S1.TIF, are supplementary figure S2, while supplementary figure S1 are in file S2.TIF. The same happened to S3 and S4, S5 and S6.

2. The mitochondrial function was not carefully evaluated, except for the measurement of ATP.

Rev. 2:

This is an interesting manuscript with extensive data that characterize the roles of Klf5 in unruptured and ruptured abdominal aortic aneurysm (AAA). They found that Klf5 is downregulated in human ruptured AAA compared to unruptured AAA. Furthermore, the authors also showed downregulation of Klf5 in mouse ruptured (42 days) compared to unruptured AAA (28 days) induced by Ang II in ApoE-/- mice. To further characterize the specific function of Klf5 in the pathogenesis of AAA, they generated mice lacking Klf5 in SMCs and showed a role of Klf5 deficiency in mouse AAA biogenesis induced by Ang II. Subsequently, the authors found that loss of Klf5 in VSMCs enhanced mitochondrial fission via upregulation of the pro-fission proteins Drp1, Fis1 and Mtfr1 and downregulation of the pro-fusion protein Mfn1. Finally, they found that mitochondrial fragmentation in VSMCs is a result from eIF5a reduction elicited by Klf5 downregulation…

The data presented in this manuscript provide new insight into how KlF5 affects the AAA biogenesis through regulating the expression of eIF5a, and in turn eIF5a regulate mitochondrial dynamics in VSMCs. However, experiments that support their conclusions need to be improved. Therefore, although promising, the manuscript does not seem suitable for publication in its present form.

The major points that must be addressed:

1. First of all, throughout the entire manuscript, the authors used mouse primary VSMCs and human VSMCs in different experiments, but they were not clearly indicated in each experiment. For example, the results in Figure 3 were obtained from human VSMCs? Figure 4C, D and F from human VSMCs? Figure 4H from mouse primary VSMCs? The same question is found in Figures 5-7. It is not clear why the authors wanted to use mouse primary VSMCs or human VSMCs in different experiments.

2. The description in the Figure Legends is unclear and should be re-written to make the figure easily understood.

3. In Figure 1A and 1B, the author analyzed the expression pattern of Klf5 protein by western blotting in patient tissues of unruptured AAA (22 cases), ruptured AAA (4 cases) and 8 control samples. However, they only showed the WB results in part of analyzed samples. It is very important to show levels of Klf5 protein in each sample including 8 controls, 22 unruptured AAA and 4 ruptured AAA. Additionally, in Figure 1C and 1D, immunohistochemistry of tissue sections that were collected from paraffin-embedded samples? and the sections came from the patient samples in Figure 1A should be indicated.

4. In Figure 1E and 1F, mock control mice infused with saline at 28 and 42 days should be added.

5. In Figure 1H, how many mice at indicated time points were analyzed by immunohistochemistry should be indicated. Similar to those analyzed in human AAA as shown in Figure 1A, the authors should assess expression of Klf5 by western blotting in mouse ruptured AAA and unruptured AAA induced by Ang II in ApoE-/- mice.

6. Given the importance of abnormal Klf5 expression in human AAA, it will be interesting to analyze the expression pattern by western blotting in mouse unruptured AAA and ruptured AAA induced by Ang II in ApoE-/- mice (at least 5 samples will be analyzed in each group).

7. In Figure 2G, the immunohistochemistry of tissue sections (paraffin-embedded) from tumor areas of mouse unruptured AAA (at 3 month) and ruptured AAA (at 18 months) or should be clearly indicated. Additionally, the immunohistochemical images in Figure 2J are too low magnification to recognized the CD45-positive cells.

8. In Figure 3A-3C, the mock controls should be added in each indicated time points (for example, at 1, 3 and 5 days). In Figure 3B and 3I, the authors should analyze Ki-67-positive cells (%) but not Ki-67 relative fluorescence.

9. In Figure 4, it is very confused to use mouse or human cells in different experiments. The authors should clearly indicate human or mouse cells used in different experiments. For examples, Figure 4A and 4B were probably performed in mouse materials, whereas Figure 4C, 4D and 4F probably in human VSMCs, and 4G and 4H used mouse VSMCs? For the experiments in Figure 4E and 4F, to compare the effect of knockout and overexpression of Klf5 on the levels of the mitochondria-shaping proteins, the authors should perform these experiments in the same species (in mouse or in human VSMCs). Therefore, the authors should evaluate expression of those proteins indicated in Figure 4E using human VSMCs depleted of Klf5 by siRNA, and repeat experiments as shown in Figure 4H to see whether knockdown and overexpression of Klf5 results in an opposite effect on mitochondrial morphology in human VSMCs. In Figure 4D, ablation of Klf5 in mouse primary VSMCs (?) changes the expression levels of Fis1, Drp1, Mtfr1, Mfn1 and Pink1. How about the expression levels of these proteins if Klf5 re-introduced into Klf5-deficient cells?

10. In Figure 5, were all the experiments performed in human cells? This should be indicated in each figure legend. The levels of Klf5 should be added in Figure 5D.

11. Again, in Figure 6, were all the experiments performed in mouse primary VSMCs or human VMSCs? This should be indicated in each figure legend. In co-immunoprecipitation assays shown in Figure 6A, the input data should be added. In Figure 6C, the author suggested that overexpression of eIF5a promotes mitochondrial fusion via an enhanced interaction with the pro-fusion protein Mfn1, whereas knockdown of eIF5a leads to an opposite effect, but they did not show the data about mitochondrial morphology analysis. Moreover, overexpression or knockdown of eIF5a seems to have an effect on levels of Mfn1 in cells as seen in Figure 6C. Thus, to better understand the potential role of eIF5a in regulating mitochondrial dynamics, the authors further evaluate the effects of overexpression or knockdown of eIF5a on the levels of Fis1, Mff, MIEF1/Mid51, MIEF2/Mid49, Drp1, Mtfr1, Mfn1, Mfn2 and OPA1 (by western blotting).

12. In Figure 8A, the authors should use scatter plots that show all the individual data points and standard deviation (SD) to represent variation of eIF5a expression in human unruptured (n=22), ruptured (n=4) AAA and adjacent control arotas (n=8).

Rev. 3:

In this manuscript, Dong Ma et al. investigate the functional relationship of KLF5 with eIF5a and their involvement in the formation and rupture of abdominal aortic aneurysm via regulation of the proliferation of vascular smooth cells and induction of oxidative stress. This is an interesting study with potential ramifications in understanding of the molecular mechanisms that underlie biology of aneurysms, which may eventually lead to therapeutic interventions. The authors performed a combination of in vivo and in vitro experiments along with analyses of human samples, which signify further the translational potential of their findings. Here are the comments of this reviewer:

1. Although it is clear that KLF5 expression is upregulated at the early stage of the AAA and its expression returns back to normal levels at the late stage, it remains elusive how the authors came up with the conclusion that KLF5 is protective in late phase, when the disease has already reached the maladaptive stage. The biphasic pattern of KLF5 expression requires pharmacologic interventions at the early vs. late stage in order to identify the critical stage that KLF5 suppression may be beneficial or detrimental. The authors can apply pharmacologic treatment (ML264) in mice treated with AngII aiming to block KLF5 expression at the early stage only vs. continuous treatment (early + late stage) and monitor progression of the disease.

2. In light of the need for investigation of the association of the biphasic alteration in KLF5 expression, the in vitro experiments of combined treatment with AngII and Ad-KLF5 or Ad-shKLF5 (Fig 3D, 3E) need to include more time points (Day 0, 1, 3, 5) as for Fig 3A-C and not only Day 0 and Day 5.

3. The authors need to demonstrate intensity of AngII signaling following long-term treatment of mice with the stimulator of AAA formation. Previous studies (e.g. Gao D. et al.; PLoS One. 2015) have shown that AT1 receptor antagonist (losartan) attenuates KLF5 expression. Thus, it is possible that the attenuation in KLF5 expression that is observed in the late stage of the disease is an effect of desensitization of the AngII receptor and not an active event of the AAA pathophysiology.

4. Oxidative stress is now presented in a way that is disconnected by the biochemical pathways that underlie it. The authors propose that Drp1-mediated fission is over-activated and leads to accumulation of fragmented mitochondria that account for oxidative stress. This presentation precludes from understanding the source of reactive oxygen species, accumulation of which is critical for the development of the AAA. The authors should investigate further the association of AngII signaling, KLF5, eIF5α with the expression of proteins that are involved in enzymatic and non-enzymatic sources of ROS.

5. Mitochondrial function assays e.g. activity of mitochondrial enzymes, Seahorse analysis, along with assessment of markers for mitochondrial number (mtDNA:nuDNA) and replication (mtTFA, PGC1 etc.) in the early and late stage of AAA development in mice and/or in vitro would help to identify whether impairment of mitochondrial health is a cause or consequence of oxidative stress and eventually a major driver for the development of the disease.

6. Treatment of mice with an anti-oxidant in the early vs. late stage of AAA would help to identify the extent of the involvement of oxidative stress in the effect of KLF5 and eIF5α inhibition in promoting the disease.

7. What is the level of expression of Mfn1 in Fig. 6F and of eIF5α in Fig. 6G?

8. What is the effect of combined AngII treatment and KLF5 or eIF5a overexpression in ATP content of VSMCs?

9. Minor: Rearrange panels of Fig 6, so that they are in order.

---

## [Decision Letter · Decision Letter 2]

9 Jun 2020

Dear Dr Ma,

Thank you for submitting your revised Research Article entitled "eIF5a reduction elicited by Klf5 downregulation leads to cell senescence  through inducing mitochondrial fission in vascular smooth muscle cells" for publication in PLOS Biology. I have now obtained advice from the three original reviewers and have discussed their comments with the Academic Editor. 

Based on the reviews, we will probably accept this manuscript for publication, assuming that you will modify the manuscript to address the remaining points raised by Reviewers 1 and 3 (see below). In addition, we would like you to consider modifying the title to improve it. We would suggest the following: "Klf5 downregulation induces vascular senescence through eIF5a depletion and mitochondrial fission."

Please also make sure to address the data and other policy-related requests noted at the end of this email.

We expect to receive your revised manuscript within two weeks. Your revisions should address the specific points made by each reviewer. In addition to the remaining revisions and before we will be able to formally accept your manuscript and consider it "in press", we also need to ensure that your article conforms to our guidelines. A member of our team will be in touch shortly with a set of requests. As we can't proceed until these requirements are met, your swift response will help prevent delays to publication.

*Copyediting*

*Published Peer Review History*

*Early Version*

*Submitting Your Revision*

Sincerely,

Ines

--

Ines Alvarez-Garcia, PhD

Senior Editor

PLOS Biology

Carlyle House, Carlyle Road

Cambridge, CB4 3DN

+44 1223–442810

ETHICS STATEMENT:

-- Thank you for specifying the name of the ethics committee that reviewed and approved the animal care and used project license. Please also include an approval number.

DATA POLICY:

Thank you for sending us the data files to comply with our data policy. Note that the numerical data provided should include all replicates AND the way in which the plotted mean and errors were derived (it should not present only the mean/average values). I have now checked the data and I found several outstanding points. Please address the following issues:

- Add all the values (not only the mean) underlying the graphs in Fig. 1B, D

- Add all the values for the graphs shown in Fig. 2B, F (note there is an error and you labelled it 2E – please correct this)

- Add all the values for the graphs shown in Fig. 3B, C, D, E, F, G, H

- Add the data for Fig. 4B and all the values for the graphs shown in Fig. 4D, G, H

- Add all the values for the graphs shown in Fig. 5B, C, D, F

- Add all the values for the graphs shown in Fig. 6D, E, F, G, I, J

- Add all the values for the graphs shown in Fig. 7A, B, C, D, H, I, J

- Add all the values for the graphs shown in Fig. S3B, C

- Add all the values for the graph shown in Fig. S4A

- Add all the values for the graph shown in Fig. S6

- Add all the values for the graphs shown in Fig. S9A, B

- Add all the values for the graphs shown in Fig. S10A, F, G, H, I

- Add all the values for the graphs shown in Fig. S12A, B, C

- Add all the values for the graphs shown in Fig. S13A, B

- Add all the values for the graph shown in Fig. S14

- Add all the values for the graphs shown in Fig. S17A, B, C, D, E, F

- Add all the values for the graphs shown in Fig. S18A, B

Reviewers’ comments

Rev. 1:

The authors have addressed my questions with new experiments and data. I have a couple of minor points:

1. Supplementary Fig. 10H and I, although these results demonstrated that AngII treatment caused mitochondria dysfunction in a time dependent manner and suggested a causal relationship between AngII and the impairment of mitochondrial funciton, the role of Klf5 cannot be inferred from these data. Therefore, the conclusion should be revised.

2. Supplementary figure 10E, the label of the 3rd column should be "smcKlf5-/-+Ad-Klf5".

Rev. 2:

In the revised manuscript, the authors have addressed my concerns and I have no additional comments.

Rev. 3:

The authors have made a significant effort to address the comments of this reviewer.

The new experiments that they performed do not support the notion that KLF5 suppression at the late stage of AAA is a protective response (Fig S2A,B). In fact, this figure should be part of the main Figures panel and not in the Supplement. Accordingly, the final version should focus primarily on the detrimental effect of KLF5 in driving AAA following activation of the protein. Statements implying that KLF5 suppression at the end stage are protective need to be removed from the text because they do not seem to reflect on the actual biological events.

The limited improvement in cells treated with an anti-oxidant (in vitro; Fig S18) and the lack of in vivo proof that suppression of oxidative stress per se can be protective limits the possibility that this is a driving force for the observed phenotype. The authors should add a "Limitations" paragraph at the end of the Discussion section stating that treatment of AAA with anti-oxidants can be only supplemental to other interventions that should target more essential pathways for the development of the disease.

---

## [Editor Report · Decision Letter 3]

31 Jul 2020

Dear Dr Ma,

On behalf of my colleagues and the Academic Editor, Cecilia W Lo , I am pleased to inform you that we will be delighted to publish your Research Article in PLOS Biology. 

Early Version

PRESS 

Kind regards,

Pamela Berkman

Publishing Editor, 

PLOS Biology

on behalf of

Ines Alvarez-Garcia,

Senior Editor

PLOS Biology